# SAETTA: high resolution 3D mapping of the total lightning activity in the Mediterranean basin over Corsica, with a focus on a MCS event

Sylvain Coquillat[1], Eric Defer[1], Pierre de Guibert[1], Dominique Lambert[1], Jean-Pierre Pinty[1], Véronique Pont[1], Serge Prieur[1], Ronald J. Thomas[2], Paul R. Krehbiel[2], William Rison[2]

[1]Laboratoire d'Aérologie, Université Toulouse 3 Paul Sabatier, CNRS, Toulouse, France
[2]Langmuir Laboratory for Atmospheric Research, New Mexico Tech, Socorro, New Mexico, USA

*Correspondence to*: Sylvain Coquillat (sylvain.coquillat@aero.obs-mip.fr)

**Abstract.** Deployed in the mountainous island of Corsica for thunderstorm monitoring purpose in the Mediterranean Basin, SAETTA is a network of 12 LMA stations (Lightning Mapping Array, designed by New Mexico Tech, USA) that allows the 3-D mapping of VHF radiation emitted by cloud discharges in the 60-66 MHz band. It works at high temporal (~ 40 ns in each 80 μs time window) and spatial (tens of meters at best) resolution within a range of about 350 km. Originally deployed in May 2014, SAETTA was commissioned during the summer and fall seasons and is now permanently operational since April 2016 until at least the end of 2020. We first evaluate the performances of SAETTA through the radial, azimuthal, and altitude errors of VHF source localization with the theoretical model of Thomas et al. (2004). We also compute on a 240 km × 240 km domain the minimum altitude at which a VHF source can be detected by at least 6 stations by taking into account the masking effect of the relief. We then report the 3-year observations on the same domain in terms of number of lightning days per square kilometer (i.e. total number of days during which lightning has been detected in a given 1 km-square pixel) and in terms of lightning days integrated across the domain. The lightning activity is first maximum in June because of daytime convection driven by solar energy input, but concentrates on a specific hot spot in July just above the intersection of the three main valleys. This hot spot is probably due to the low-level convergence of moist air fluxes from sea breezes channeled by the three valleys. Lightning activity increases again in September due to numerous small thunderstorms above the sea and to some high precipitation events. Finally we report lightning observations of unusual high altitude discharges associated with the mesoscale convective system of June 8, 2015. Most of them are small discharges on top of an intense convective core during convective surges. They are considered in the flash classification of Thomas et al. (2003) as small-isolated and short-isolated flashes. The other high altitude discharges, much less numerous, are long range flashes that develop through the stratiform region and suddenly undergo upward propagations towards an uppermost thin layer of charge. This latter observation is apparently consistent with the recent conceptual model of Dye and Bansemer (2019) that explains such upper level layer of charge in the stratiform region by the development of a non-riming ice collisional charging in a mesoscale updraft.

## 1 Introduction

Lightning is a multiscale phenomenon that occurs at the end of a chain of dynamical and microphysical processes that act throughout the formation and the lifetime of a thunderstorm cloud. The key processes are low level convergence of moist air, convection, liquid and solid condensation of water vapor, latent heat release, interactions between cloud particles, precipitation, all resulting in cloud electrification. Lightning acts as a relaxation and a limiter of the electric field resulting from the cloud electrification due to microphysical interactions especially by the non inductive charging process during graupel - ice crystals collisions (see Saunders, 2008). It can therefore be considered as a bulk tracer of the intense convection, or as a proxy of each of the cloud processes mentioned above provided it is observed at high spatial and time resolutions.

Various lightning detection systems are used so far in operational mode or for scientific research purpose. Most of them are designed to detect the electromagnetic field radiated by lightning in the Low Frequency (LF; 30 kHz – 300 kHz) and Very Low Frequency (VLF; 3 kHz - 30 kHz) ranges. Those ones (e.g. Euclid-Météorage, ATDnet, ZEUS in Europe; NLDN in USA; or WWLLN worldwide) can localize at rather long range the ground impact of the return stroke that is the most powerful phase of a cloud-to-ground lightning flash (CG), or 2D localize intracloud discharges. They do not provide yet comprehensive information on the thunderstorm cell by which they are produced. Some detection systems such as the LMA (Lightning Mapping Array of the New Mexico Tech, see Thomas et al., 2004) are designed to observe lightning in the Very High Frequency (VHF; 30 MHz – 300 MHz) range, which allows detecting the lightning leader phases and mapping the whole lightning branches inside the cloud. This provides information on lightning closely linked with the microphysical structure of the cloud in which the discharges propagate.

In the frame of the HyMeX program (Ducrocq et al., 2014), which aims at documenting the water cycle in the Mediterranean basin that is considered as a climatic hot spot (Giorgi, 2006), we plan to monitor the convection for addressing the question about the evolution of high precipitation events and deep convection in response to the climate change. For this purpose, the Collectivité Territoriale de Corse - via the PCOA (CORSiCA Atmospheric Observations Platform, https://corsica.obs-mip.fr/) - gave us the opportunity to equip the Corsica Island with a LMA network that is considered as a reference for the accurate detection of total lightning activity. The instrument is called SAETTA, which name is the acronym of "Suivi de l'Activité Electrique Tridimensionnelle Totale de l'Atmosphère" (Monitoring of the Total Tridimensional Electrical Activity of the Atmosphere) that means "lightning" in Corsican language. In combination with the CG observation of Météorage (the French part of the Euclid network), this setup provides a comprehensive description of the total lightning activity.

As a matter of fact, Corsica has a complex and tortuous relief, with mountainous massifs made up of more than a hundred summits, culminating at more than 2000 meters altitude, located only a few kilometers from the coasts. So much so that it is

often described as "a mountain in the sea". Another important aspect is the upwind sea surface evaporation forced by synoptic flows (Adler et al., 2016; Scheffknecht et al, 2016). These characteristic features of mountainous island explain the torrential nature of the rivers of Corsica. The climatic particularities related to its geographical position combined with its specific relief generate very violent precipitations which can spread over time, generating torrential floods especially in autumn (Lambert and Argence, 2008; Scheffknecht et al., 2016; Scheffknecht et al., 2017). The deployment of the LMA in this area will allow us to address scientific issues related to stormy convection in a complex maritime and mountain environment, where fine scale processes make its forecast trickier. Sea, valley and slope breezes are expected to play a key part in the triggering of convection (Barthlott and Kirshbaum, 2013; Tidiga et al., 2018), the influence of mid- and upper-level synoptic fluxes on these local dynamical features can even complicate the whole (Ducrocq et al., 2008); meanwhile low level fluxes are continuously carrying humid air from the surrounding sea (Adler et al., 2016).

The initial deployment of SAETTA in Corsica took place in May 2014. SAETTA functioned operationally from July to October in 2014, from April to December in 2015, and has been in permanent operation since April 2016 until at least the end of 2020. It is planned to operate well beyond in order to obtain long-term observations for issues related to climatic trends. So far, SAETTA has documented lightning activity from the regional scale to the flash scale, providing a monthly climatology showing different trends from one month to the next, but also specific observations of lightning of all types - including unusual flashes showing a jump towards the top of the cloud in its trailing stratiform region - and even inverted dipolar structures.

In the following, we first present the SAETTA network and its performances in Section 2, then describe the overall observations performed from 2014 to 2016 in Section 3, bring an insight on specific and unusual events detected so far at storm and lightning scales in Section 4, and discuss the perspective of such an instrument with respect to the scientific questions addressed and to the operational needs for ground based and space-born lightning observations. Several results presented benefited from the use of the XLMA software developed by Ron Thomas (Thomas et al., 2003), which was used as an analysis and display tool.

## 2 The SAETTA Network

The SAETTA network consists of 12 LMA stations (Lightning Mapping Array, developed by New Mexico Tech, USA), which allow mapping lightning flashes in 3 dimensions in real time, at high temporal and spatial resolutions, within a range of about 350 km centered on Corsica. Actually each station independently detects - in the 60-66 MHz bandwidth - the impulsive radio frequency radiations produced by the leader phase of lightning flashes, and accurately measures the time of arrival of the signals thanks to an accurate time base provided by a GPS receiver. Hence, a leader segment - so-called VHF source hereafter - that emits an impulsive radiation from the position ($x$, $y$, $z$) and at the time t detected by at least 4 stations

of the network can be fairly accurately located and dated using the Time Of Arrival (TOA) technique (see Appendix A in Thomas et al., 2004). As a matter of fact, a minimum of 6 stations is required in the data processing although only 4 unknowns ($x$, $y$, $z$, $t$) are to be determined for each VHF source. This consideration of redundant measurements is useful for checking the solution's validity.

Several advantages arise from a 12-station network. For instance, the redundancy/reliability in case of short-term and long-term failures; the effect of localized high-rate storms on a sensor's contribution to more-distant activity; the improved geometry for geo-location of distant lightning while maintaining height accuracy for nearby low-altitude lightning channels... Another advantage would be that more VHF sources can be located during a given discharges because 2 different sets of 6 stations can detect sources in the same time window. Nevertheless, during high flash rate period or during spread and very active events each lightning is logically less well sampled since sources emitted by numerous different lightning compete to be detected by the 2 sets of 6 stations in the same time windows.

A comprehensive description of the operation of LMA stations is available in Appendix A of Thomas et al. (2004). As in all LMA systems, each SAETTA station is configured to record on internal disk the amplitude and the arrival time of the strongest radiation event - above an adjusted detection threshold - which it detects and digitizes in each time window of 80 µs. The accurate time of arrival is obtained thanks to a GPS receiver (timing error of about 12 ns) that controls the frequency of a 25 MHz oscillator allowing the data acquisition with a theoretical 40 ns time resolution within each 80 µs time window (see Thomas et al. 2004 for explanation about the actual time resolution). The data is collected on site at each station by changing the internal disks, which are brought back to the Laboratoire d'Aérologie in Toulouse (France) to implement the calculation of the 3D position of the VHF sources. In addition, each station is connected by wireless communication links via a modem and a GSM antenna in order to (i) monitor and control the station operation by displaying a large amount of information (e. g. the detection threshold, the filling rate of the internal disk, the battery voltage, the load current from the solar panel...), and (ii) to send decimated data in real time (temporal resolution degraded to 400 µs and with higher detection threshold) to a central calculator for real-time processing and display, with about 1 minute of delay (http://lma.aero.obs-mip.fr/temps_reel.html). During specific periods such as a measurement campaign, the time window of the recorded data can be reduced to 10 µs and the detection threshold of the decimated data sent by telephony can be reduced too, both remotely. The main advantage of reducing the time window is to allow detecting more VHF sources during fast lightning processes like, for example, dart leaders that typically last only a few hundred microseconds and therefore are not well sampled with an 80 µs time window.

## 2.1 Location of the SAETTA stations

The 12 SAETTA stations spread over an area about 70 km in west-east direction and about 180 km in south-north direction (see Fig. 1). The distances between two stations vary from 20.7 km to 180.8 km, with an average value of 67.3 km. This

geographic configuration is unique to SAETTA network compared to theother LMA networks, i. e. SAETTA is a relatively large network for an almost same number of stations. By comparison, the New Mexico Tech LMA network (see Thomas et al., 2004) is constituted of 13 stations with a minimum distance between them of about 12 km, a maximum of about 76 km and an average of 36 km, which are approximately half of SAETTA characteristics.

This configuration dates back to 2016. In the previous two years 2014 and 2015, the stations now located in Ersa and in Pertusato (the northen and southern ends of the network, respectively), were located in Foce di Bilia on a hill 35 km north-west of Pertusato station, and in Pinarellu on the roof of the Genoese tower of Pinarellu Island, 27 km south-east of Coscione station. This change was made because the former sites had not been entirely satisfactory in terms of noise level and

10    functionality. In this previous configuration, the distances between two stations varied from 20.7 km to 118.0 km, with an average value of 59.8 km. Thus, the new configuration has led to an extension of the network in the south-north direction.

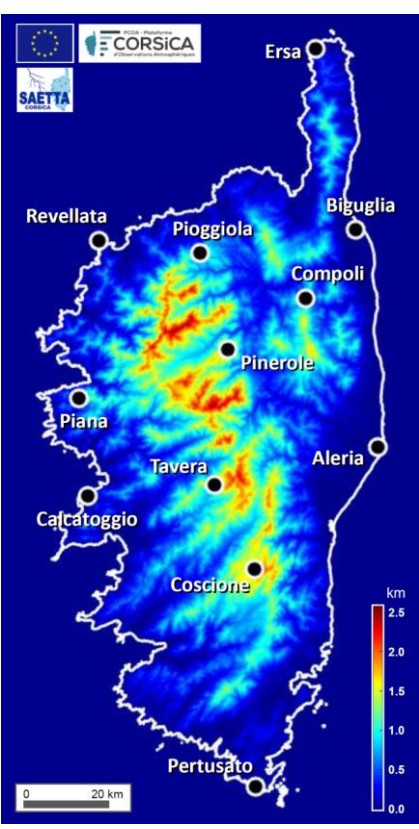

Figure 1: Map of Corsica Island with the location of the 12 SAETTA stations since 2016

The stations are located on sites as remote as possible from any electromagnetic pollution, with the widest possible field of view of the sky (LMA is line-of-sight, so only detects sources above the horizon), and with proper access to the GSM mobile phone network. The choice of sites had to face two challenges: (i) put as many stations aloft to maximize the power of detection by avoiding any masking effect by the relief, and (ii) find accessible sites, these two objectives being most often antithetical. Basically 5 stations are located on the summit of relatively high mountains along the main south-north dorsal ridge; the rest are installed on the outskirts of Corsica, more or less close to the sea in relatively high and unobstructed places. This configuration makes it possible to detect VHF sources at low altitude by at least 6 stations on either side of the central mountain range of the island. It nevertheless has a disadvantage in winter conditions since the altitude stations can be covered with snow and automatically put into hibernation during this period.

| Site | Site name | Altitude AMSL (m) | |
| --- | --- | --- | --- |
| | | 2014 – 2015 | 2016 – today |
| A | Biguglia | 3.8 | 3.8 |
| B | Aleria | 36.4 | 36.4 |
| C | Pioggiola | 1281.4 | 1281.4 |
| D | Revellata | 162.0 | |
| E | Calcatoggio | 346.2 | 346.2 |
| F | Foce di Bilia | 551.4 | |
| G | Piana | 823.2 | 823.2 |
| H | Tavera | 1648.2 | 1648.2 |
| I | Compoli | 1237.9 | 1237.9 |
| J | Pinerole | 1950.2 | 1950.2 |
| K | Coscione | 1746.8 | 1746.8 |
| L | Pinarellu | 65.3 | |
| M | Pertusato | | 104.1 |
| N | Ersa | | 357.5 |
| O | Revellata | | 167.2 |
| | Vertical difference (m) | 1946.4 | 1946.4 |

Table 1. Altitude AMSL (m) of the 12 SAETTA stations during the first period from 2014 to 2015 (third column) and since 2016 when 3 stations were moved (forth column). The vertical difference is the difference in altitude between the highest and the lowest stations.

Consequently, the SAETTA network has this other unique configuration: the altitude range of the SAETTA stations (see Table 1) is much wider than that of most other networks, particularly in the USA. The gap between the maximum and the minimum altitudes reaches 1946.4 m high. By comparison, the equivalent gap is less than about 520 m for the Oklahoma LMA; less than about 460 m for the New Mexico Tech LMA used during the STEPS campaign in Colorado and Arkansas; and equal to 335.4 m for the North Alabama LMA (see Koshak et al., 2004). The SAETTA configuration possesses thus a significant vertical baseline. According to Thomas et al (2004) and Koshak et al. (2004), this characteristic is expected to allow for a better determination of the VHF sources altitude for distant source. This point is addressed here after in the section 2.3.

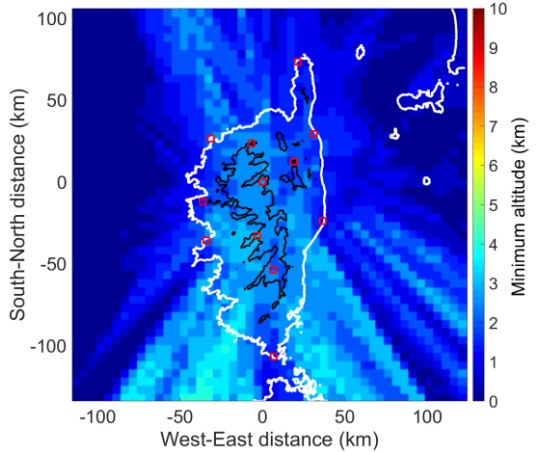 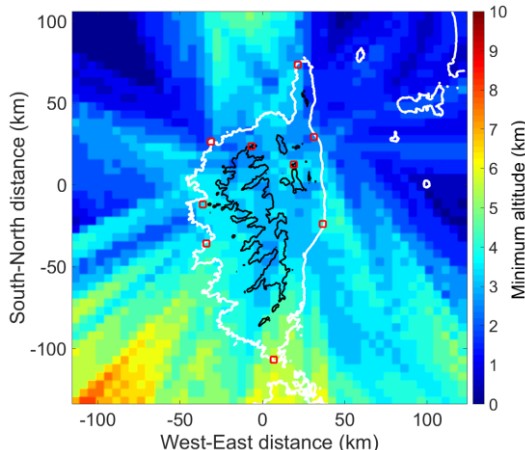

Figure 2: Capability of VHF sources detection by the SAETTA network in the 240 km × 240 km domain of better detection accuracy centered on Corsica: minimum altitude of direct vision by at least 6 stations of the network. Left: full network with 12 stations (red squares). Right: network with the 3 highest stations off (e. g. because of stations wintering). Isocontours at 1000 m altitude are indicated by black lines. Horizontal resolution: 5 km; vertical resolution: 500 m.

## 2.2 Geometric capability of VHF sources detection by SAETTA

In order to give an idea of the capability of VHF sources detection by SAETTA, the minimum altitude at which a VHF source can be directly seen by at least 6 SAETTA stations was computed, with the complete network on one hand, and with the 3 highest altitude stations off (because of snow cover in winter conditions) on the other hand. The details of the calculation, which takes into account the roundness of the Earth but not the atmospheric refraction, are presented in Appendix A. The results are displayed as maps in Fig. 2. The considered domain is 240 km × 240 km centered on the Pinerole station, which is the highest of the network (1950 m). This domain corresponds to the field of better detection accuracy by SAETTA (see Section 2.3). The chosen horizontal (5 km) and the vertical (500 m) resolutions are rather weak because the calculations are very time-consuming, but the maps give a good idea of the SAETTA detection capability. Most

of the VHF sources can be detected above about 1 km or less on the west and east sides of the island, above 2 to 3 km over and north of Corsica, and only above 4 to 5 km beyond 100 km from the center of the island in 7 sectors in the south and south-east of Corsica. In winter conditions, the detection capability deteriorates significantly especially in the south-west where it is impossible to detect sources below 7-8 km. Anywhere else, 5 km seems to be average minimum altitudes in

winter, except in the NW and NE corners of the sea domain where low levels can still be scrutinized. For longer range observation up to about 350 km from the center of the network, see section 3.1. One must keep in mind that atmospheric refraction is not taken into account in the calculations, consequently the here calculated altitudes overestimate the real minimum altitudes of VHF source detection. Actually, electromagnetic waves propagating in the clear sky are deflected downwards because of the refractive index gradient, which is most often downward directed. Therefore, VHF sources can be

detected even below the limits indicated in Figure 2.

## 2.3 Location accuracy of the SAETTA network

With regard to the uncertainty of localization of VHF sources by the SAETTA network, reference can be made to the article by Thomas et al. (2004) who evaluated both theoretically and experimentally the location accuracy of the LMA used during the STEPS 2000 experiment (Lang et al., 2004). They found that short-duration pulses emitted by a VHF transmitter carried

by a sounding balloon between 6 and 12 km altitude over the central part of the network were located with an accuracy of about 6-12 m in horizontal position and about 20-30 m in height, in the optimal situation. They also developed a geometrical model the results of which were in good agreement with experimentally observed errors from a sounding balloon and from aircraft tracks. Koshak et al. (2004) also addressed the problem of location accuracy of the LMA by developing a source retrieval algorithm for theoretically studying the location errors. More recently, Chmielewski and Bruning (2016) have

explored location errors and detection efficiency of various LMA networks in the United States by means of model simulations based on methods previously developed by Koshak et al. (2004) and Thomas et al. (2004).

The geometric model of Thomas et al. (2004) is therefore a suitable tool for evaluating the SAETTA network performances. It provides analytical formulations of the increase in the sources location uncertainties with distance, based on the spherical

coordinates ($r$, $\theta$, $\varphi$) of a VHF source relative to the center of the network. For example, the azimuth angle $\varphi$ is determined primarily by stations having the greatest separation transverse to the propagation line of the signal emitted by the sources. Therefore, the more extended the network is in any direction, the better the determination of the azimuth angle $\varphi$. Given the larger extension of SAETTA network in the south-north direction, azimuth angles are therefore expected to be better determined in the west-east direction. This is confirmed by the errors calculated with the geometrical model, for VHF

sources at 10 km altitude seen from 12 stations, and displayed in Fig. 3. The azimuthal errors, which correspond to the central graphs, exhibit lower values in the west-east direction. Furthermore, the comparison between the 2014-2015 network and the 2016 network configurations confirms that the larger the south to north extension of the network, the lower the errors are in the west-east direction. The same behavior can be pointed out from Fig. 3 for the range error since it also depends on

the transverse extent according to Thomas et al (2004). One can turn his attention to the fact that the theoretical errors are here calculated for sources located at 10 km altitude and detected by 12 stations. This is a best case useful to compare networks. But if we consider lower sources, the errors will increase, especially at low level where sources cannot be detected by so numerous stations because of the masking effect of the relief.

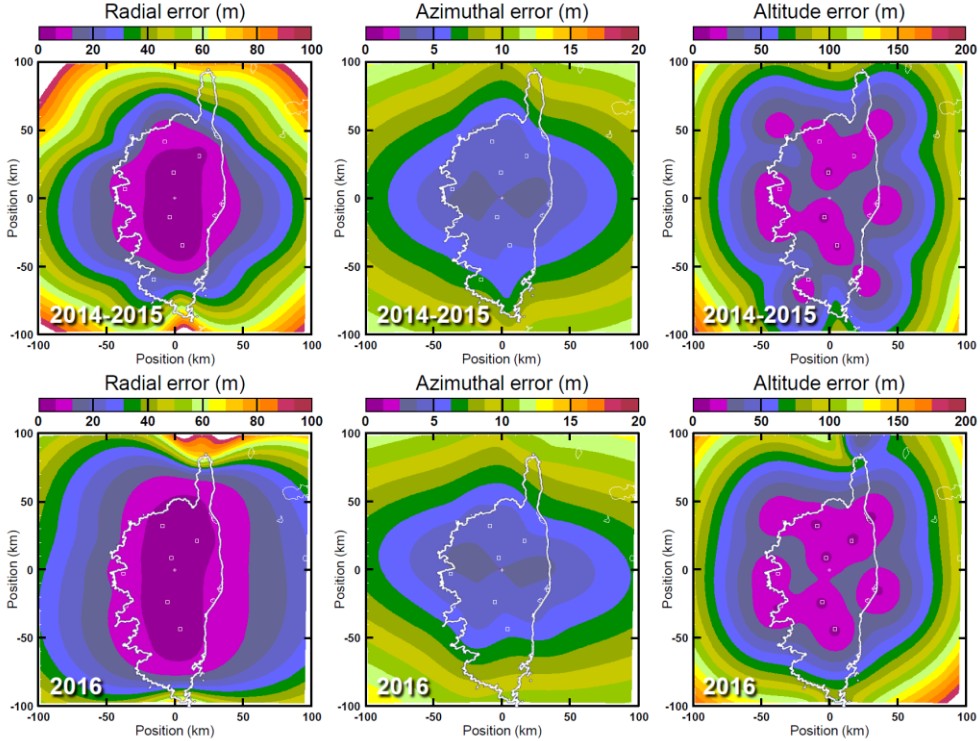

Figure 3: Radial, azimuthal and altitude errors computed with the geometrical model of Thomas et al. (2004) for VHF sources located at 10-km altitude seen from 12 stations. Top: initial SAETTA network in 2014 and 2015; bottom: new SAETTA network since 2016. Stations location is indicated by white squares.

The geometrical model also predicts that the slant range $r$ and altitude $z$ uncertainties increase versus $r^2$ while the azimuth uncertainty more slowly increases versus $r$ (see Figure 12 in Thomas et al., 2004). This behavior is illustrated in Fig. 3: far from the network, the geographical error gradient is much higher for range and altitude determination (left and right graphs, respectively) compared to the relatively small geographical error gradient of azimuth determination (central graphs). The azimuth is best determined and the corresponding errors are the smallest. Looking into detail of Fig. 3 for the 2016 network, the theoretical errors can be evaluated, for example, about 50 km from the center of the network and compared with that of the STEPS network according to Fig. 12 of Thomas et al. (2004). According to Fig. 3, the radial, azimuthal, and 10-km

altitude errors at 50 km from the center of the SAETTA network are about 15 m, 8 m, and 40 m, respectively. According to Fig. 12 of Thomas et al., the radial, azimuthal, and altitude errors at 50 km from the center of the STEPS network are about 100 m, 16 m, and 80 m, respectively (theses values are the average values of the errors in both east-west and north-south directions, i. e. black and red solid lines at the abscissa 50 km). The comparison shows that theoretical errors seem less important for the SAETTA network, especially for the slant range r. The greater horizontal extension $D$ of the SAETTA network is undoubtedly at the origin of this behavior since the theoretical errors are inversely proportional to D according to Thomas et al (2004), and even inversely proportional to the square of $D$ for the radial error.

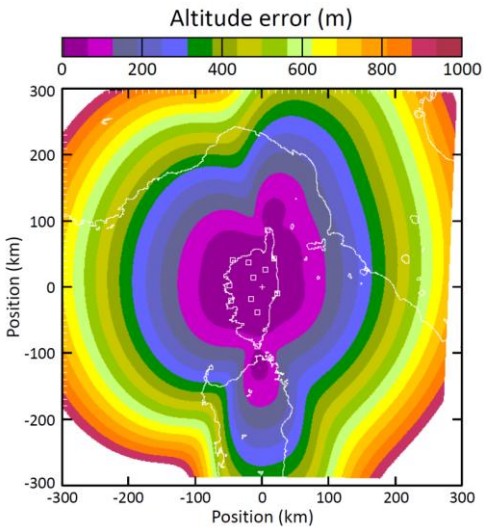 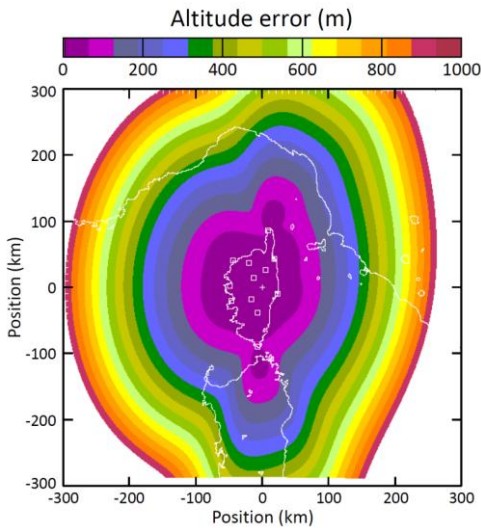

Figure 4: Altitude error computed with the geometrical model of Thomas et al. (2004) for VHF sources at 10 km of altitude seen from 12 stations in a large domain: analysis for the 2016 network (left) and for the same network considered flat with a an average altitude of about 863 m (right).

The SAETTA network possesses a significant vertical baseline that allows overcoming the insufficient vertical separation among the stations that can be source of error in the determination of the VHF sources altitude via the elevation angle $\theta$, especially for distant sources (Thomas et al, 2004; Koshak et al., 2004). In order to evaluate the contribution of this vertical baseline to the localization accuracy of VHF sources, the calculation of the altitude error was first carried out for the SAETTA 2016 network over a wide domain (300 km × 300 km), and in a second time for the same network but considered flat with all the stations located at the same average altitude of 863 m. Displayed in Fig. 4, the results confirm the above expectation for distant sources. Within a radius of about 150 km from the center of the network, the altitude errors are quite similar for both network configurations (actual and flat networks). The main differences arise in regions beyond this radius,

where the altitude error strongly increases for a flat network. Therefore, the vertical baseline due to having some stations at higher altitudes improves the accuracy for lightning detection over the mainland of Italy and France.

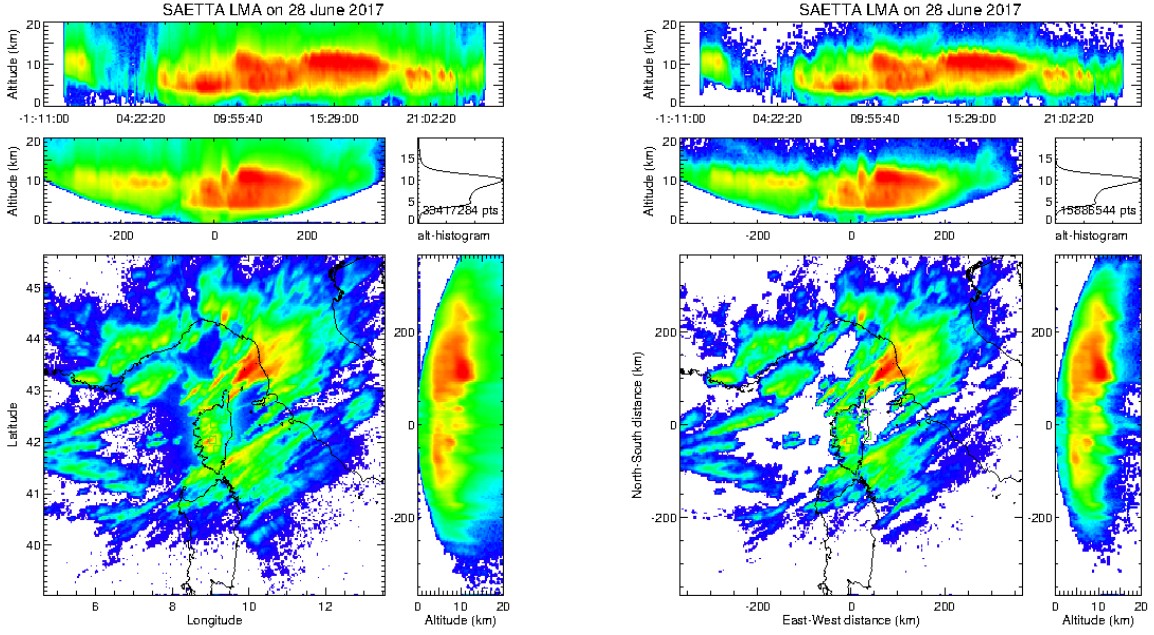

Figure 5: VHF sources detected by SAETTA during the storm event of 28 June 2017. Left: sources density of unfiltered data with angular projection (minimum of 6 stations, $\chi^2 \leq 5$). Right: sources density of filtered data with Cartesian coordinate projection on the same domain (minimum of 7 stations, $\chi^2 \leq 0.5$). Composition of each graph: altitude vs. time (top); altitude vs. longitude or view from the south (center left); VHF sources altitude histogram (center right); latitude vs. longitude or geographical projection (bottom left); altitude vs. latitude or view from the west (bottom right).

## 3 Overall observation from 2014 to 2016

### 3.1 Typical lightning observation

SAETTA is able to detect flash activity up to approximately 350 km from the center of the network (Pinerole station). An example of this large scale detection is displayed in Fig. 5 as VHF sources density for the 28 June 2017 event. The left graph corresponds to minimally filtered data (minimum of 6 stations and reduced chi-square $\chi^2 \leq 5$, see Thomas et al., 2004). It is composed of various panels according to the conventional XLMA format: altitude vs. time (top panel); altitude vs. longitude or view from the south (center left panel); VHF sources altitude histogram (center right panel); latitude vs. longitude or geographical projection (bottom left panel); altitude vs. latitude or view from the west (bottom right panel). One can see on

the vertical projections (center left and bottom right panels) that lower layers are poorly and even not documented far from the network because of the Earth roundness, as it is with meteorological radars. Unfortunately, locally generated noise events can lie close in time to lightning events and therefore may be considered in the calculation of VHF sources position, producing slightly fuzzy lightning contours. The best way to minimize their impact on observations is to restrict the sources

to those located by more than 6 stations with a better goodness of fit (i. e. lower reduced chi-square). The right graph in Fig. 5 corresponds to such filtered data: the VHF sources have been determined with a minimum of 7 stations and with $\chi^2 \leq 0.5$. Those values were determined empirically for SAETTA based on several observations at different time and space scales. By comparing the geographical projections (bottom left panels) of left and right graphs in Fig. 5, one can see that the highest densities remain almost not affected by the filtering. The same behavior is visible at lightning scale (not shown here) where

discharge channels remain very well described and noisy sources are eliminated after filtering. From here on, all the results presented below will relate to VHF sources determined with this filtering (minimum of 7 stations and $\chi^2 \leq 0.5$).

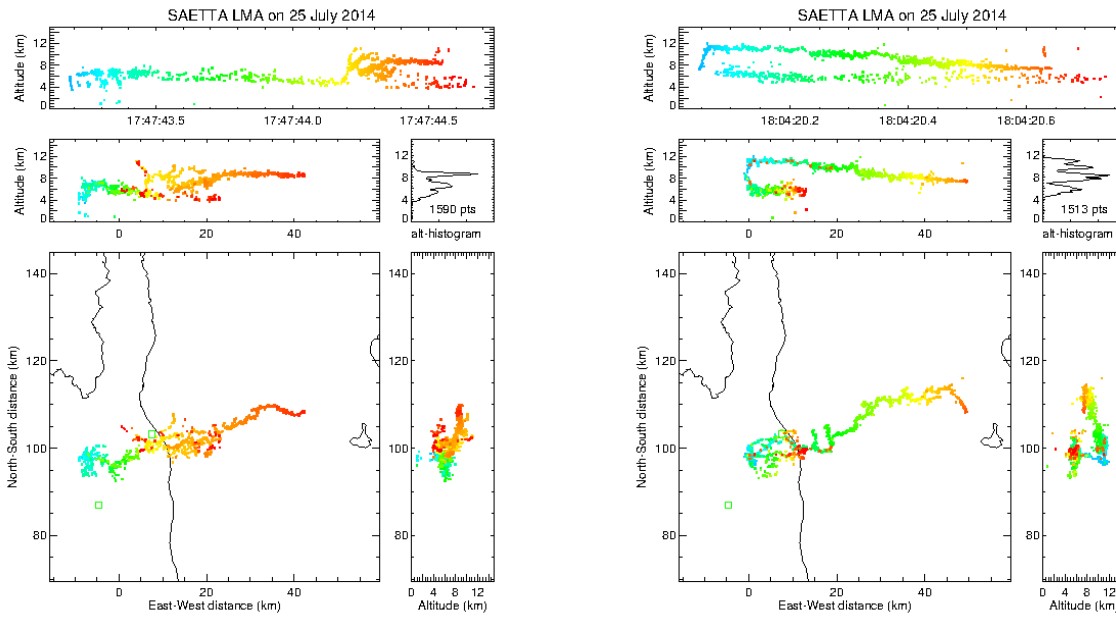

Figure 6: Examples of CG (left) and IC (right) lightning detected by SAETTA on 25 July 2014. Each dot corresponds to a

VHF source detected by SAETTA. The colors indicate the time during the sequence: from blue (beginning of the time window) to red (end of the time window). Stations location is indicated by green squares. Composition of each graph: same as in Fig. 5.

Another example of SAETTA detection is displayed in Fig. 6. Here the observation is made at the lightning scale during about 1 second. The left graph represents a Cloud-to-Ground (CG) lightning. Looking to the top frame (altitude of VHF sources versus time), one can easily see that the initial phase of the discharge (beginning of the time sequence in blue color) consists of a descending negative leader that starts at about 6 km high just before 14:47:43.2 and descends to about 3.6 km after which it is no longer reported by LMA. Its propagation speed is roughly evaluated at $3 \times 10^5$ m s$^{-1}$. It is followed by four other leaders propagating towards the ground between 14:47:43.2 and 14:47:43.4. The polarity of the leaders is deduced from the power with which the VHF sources are detected since intermittent negative leaders radiate much more than continuous positive leaders. Afterward a positive leader propagates almost horizontally over about 15 km in the trailing stratiform region of the cloud according to the plan view in the bottom left panel until 17:47:44.2 when a negative leader suddenly propagates upward to the upper part of the cloud and spreads over about 40 km. In the mean time, a positive leader subsequently appears to develop in the lower layer at about 4.5 km high. It looks like an IC flash started at 17:47:44.2, finally producing a hybrid flash. This flash is very similar to the event M reported by van der Velde and Montanya (2013) in their Figure 6. A time-distance plot could provide a good estimation of the speed of ascent of the upper negative leader, so that one can refer to the time-distance plot of the M event in the Figure 7 of van der Velde and Montanya (2013) to have an idea about the scale of speed values.

The right graph in Fig. 6 represents an Intra-Cloud (IC) lightning flash. The corresponding top panel shows that the initial phase of the discharge corresponds to an ascending negative leader triggered at about 7 km high and propagating up to about 11 km. Then it propagates over approximately 55 km following a slow descent, which is most probably due to the sedimentation of the charged ice particles that have been transported in the stratiform region and on which the discharge connects (Carey et al., 2005; Ely et al., 2008). About 30 ms after the triggering of the negative leader, the positive leader propagates also horizontally in the lower part of the discharge. In the end of the time sequence (red color, at about 12:04:20.63) a fast ascending discharge (recoil leader) follows the main vertical branching of the flash then propagates almost horizontally in the upper part of the initial negative leader path (spaced red dots in the center left panel).

## 3.2 Short lightning climatology over Corsica

In order to assess the behavior of convection over Corsica, overall data for all years of observation are accumulated. The data analysis focuses on the 240 km × 240 km domain of better detection accuracy centered on Corsica with filtered data (VHF sources determined with a minimum of 7 stations and with a reduced-chi square $\chi^2 \leq 0.5$). At the present time the data processing and analysis of 3 years of observation have been completed (2014; 2015; 2016). The sample size is not yet large enough to call this Climatology but it gives the first tendencies of the stormy behavior in this region.

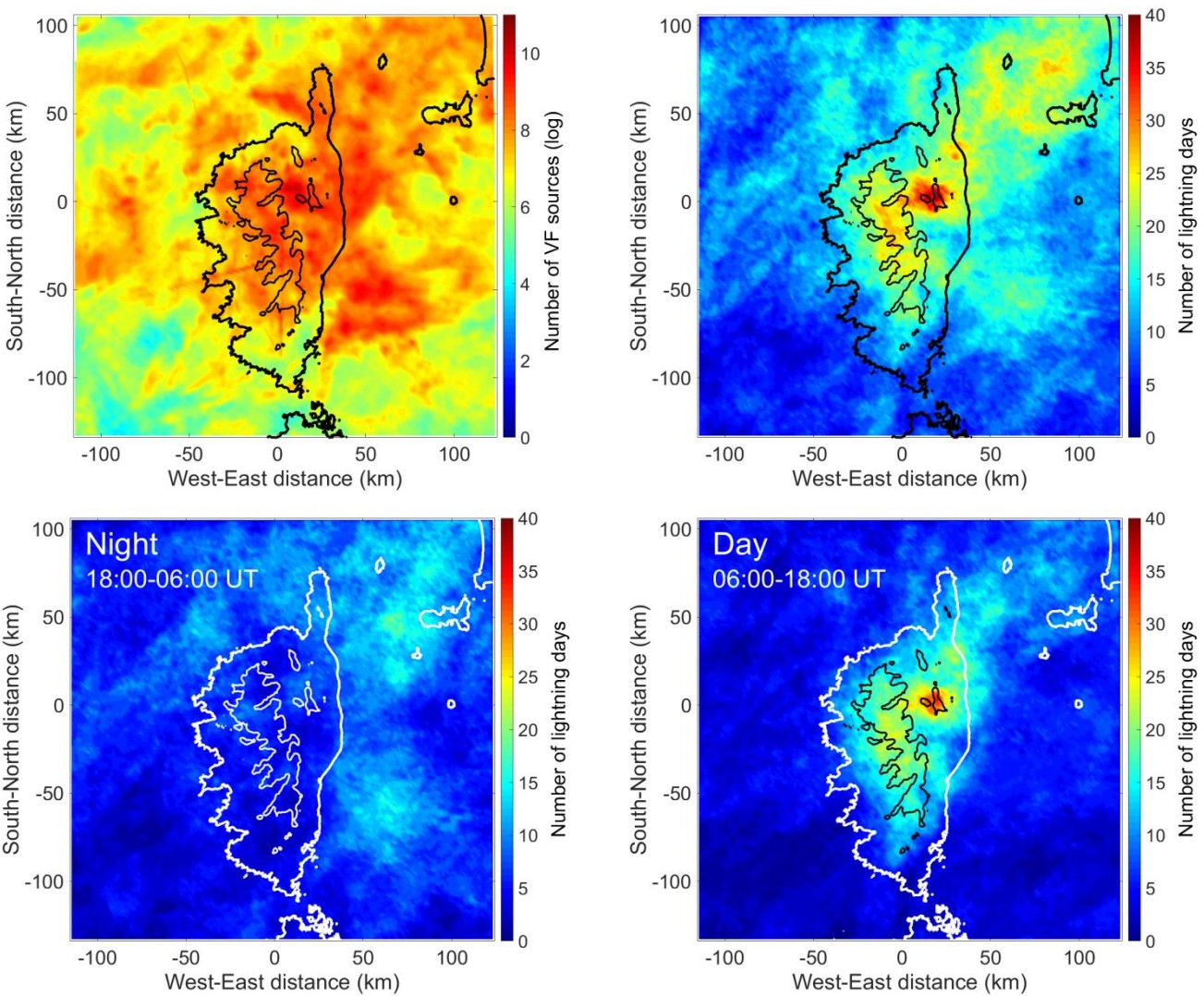

Figure 7: Overall cumulative number of filtered VHF sources in the 240 km × 240 km domain of better detection accuracy centered on Corsica for 2014 to 2016 years (top left); frequency of lightning days for the overall set of filtered data on the same domain and during the same period (top right); frequency of lightning days at night (bottom left) and day (bottom right). A lightning day in a given pixel is defined as a day during which at least 5 VHF sources are detected in that pixel. Isocontours at 1000 m altitude are indicated by black or white lines inside Corsica.

The overall cumulative number of filtered VHF sources is displayed in Figure 7 (top left). The VHF sources are counted on 1 km × 1 km pixels. Not clearly visible in this top left figure with a log scale in base 10, the highest values (dark red) are located above the main relief of the island of Corsica, and more specifically in the center of the northern part at the

crossroads of 3 large valleys, roughly between the Pinerole and Compoli stations (see Fig. 1). Other high values are located above the sea near the east coast, and also west of Cape Corse (long relief oriented south-north, forming the northern tip of Corsica).

We built a "storm days" map by counting the total number of days during which at least 5 VHF sources have been detected, in each 1 km × 1 km pixels. This value of 5 sources has been tested and chosen so as not to take into account isolated sources corresponding to residual noise or poorly located sources, while keeping the events with low lightning activity. It is a compromise that respects the storm activity actually observed. The overall frequency of storm days for 2014 to 2016 years is displayed in Fig. 7 (top right). The high values patterns are somewhat different from that of the cumulative number of VHF

sources. A big maximum (dark red, 10 km south of Compoli station) appears in the northern part of the island few kilometers to the east of the maximum of VHF sources displayed in Fig. 7 (top left). Secondary maxima (orange and yellow) are located over the central main relief; over a small area close to the Biguglia station (see Fig. 1); and over the sea in the north-east quarter of the domain. The high values of VHF sources number (Fig. 7 on the top left) corresponding to low frequencies (Fig. 7 on the top right) are obviously the signature of very intense events, e. g. almost everywhere over the sea where red

color is displayed in the top left graph of Fig. 7.

By comparing the 2 graphs at the top of Fig. 7, one realizes that the north center perimeter encompassing the crossroads of the 3 main valleys and the relief south of the Compoli station seems to be a place where thunderstorms are together very active and often present. The question that comes naturally to mind is when does it happen and why does it happen in this

place? To answer to the first point, the study of the frequency of the events was differentiated on the one hand between night and day and on the other hand according to the month of the year. The frequency of lightning day during night between 18h and 6h UT is displayed in Fig. 7 (bottom left) and that during day between 6h and 18h UT in Fig. 7 (bottom right), both with the same color scale as the overall frequency (Fig. 7, top right). It is clear that night is much less affected by convection and when it is, it occurs rather over the sea meanwhile day lightning activity provides the main contribution to the overall

frequency pattern. A more precise study of the diurnal evolution of the lightning activity was carried out for the observation of July only. The results not presented here show that the maximum activity is due to storm events occurring between 11:00 UT and 14:00 UT. As expected, the maximum of lightning frequency over the relief of Corsica is thus mainly due to diurnal convection.

The differentiation of the storm frequency according to the month of the year is displayed in Fig. 8 from June to September (most active months). June is characterized by a maximum of lightning activity over the whole relief, and especially in the center of the island and just south of Compoli station. In July the most frequent lightning activity concentrates over the northern half of the island, with a main maximum close to the crossroads of the 3 main valleys, and a secondary maximum in the north-east quarter of the domain in the Gulf of Genoa. In August, the lightning activity becomes scarce with a maximum

located close to the east coast in the vicinity of the Biguglia station (see Fig. 1) and with again a secondary maximum in the Gulf of Genoa. At last in September the maximum lightning activity is located over the sea 20-40 kilometers east of Cap Corse, while secondary maxima are present on the west flank of southeast main relief and close to the southeast coast. The maximum of lightning day frequency observed in Fig. 7 is thus mainly due to July and in some extent to June lightning activities.

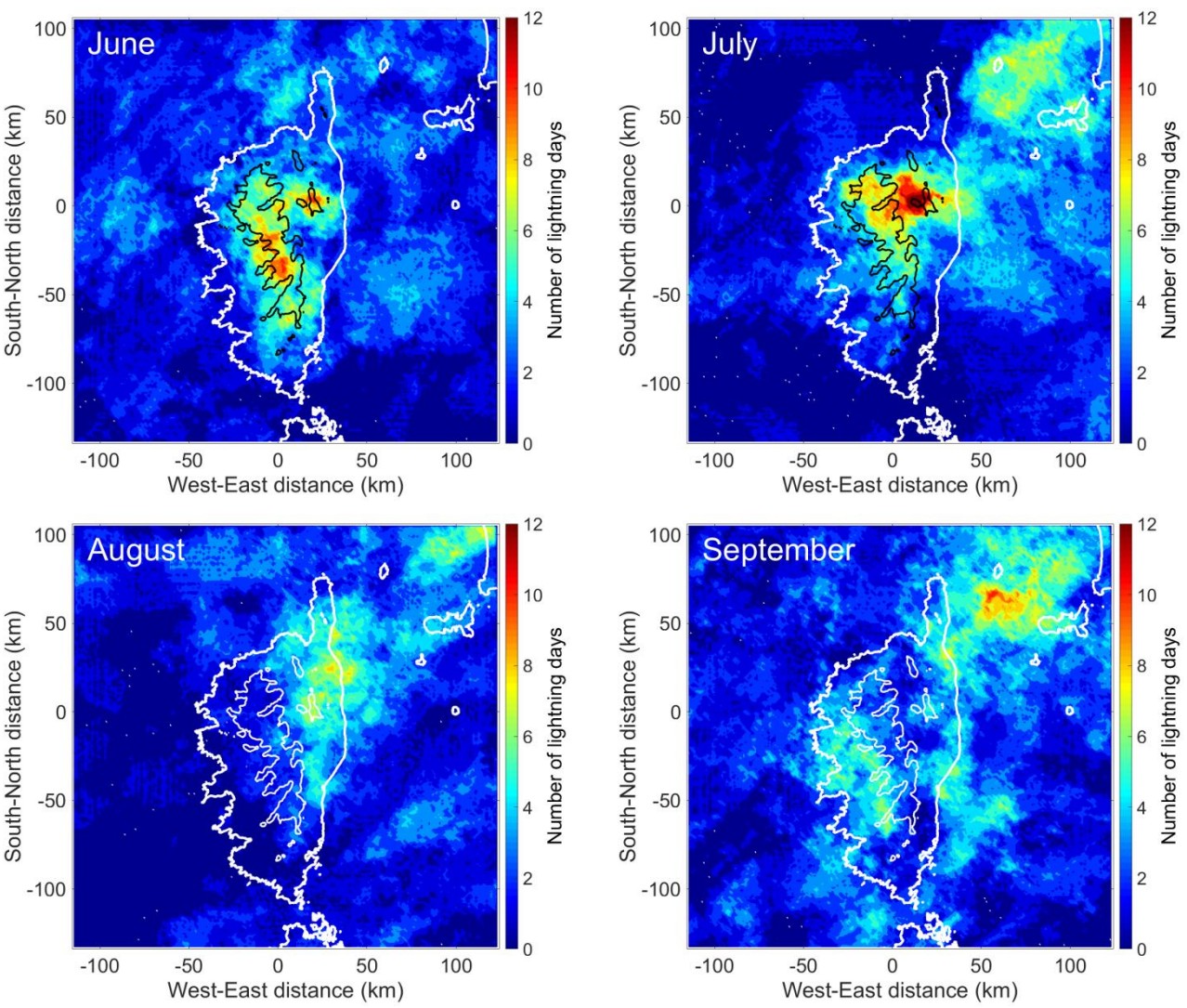

Figure 8: Maps of monthly number of lightning days in the 240 km × 240 km domain of better detection accuracy centered on Corsica from June to September for 2014 to 2016 years. Isocontours at 1000 m altitude are indicated by black or white lines inside Corsica.

To complete these geographical observations, one can also look at the overall observations reported as a histogram in Fig. 9. This figure shows the total number of lightning days on the domain of better detection accuracy for each month and each year, together with the monthly average number of lightning days. The available data set is here enriched with the dates of the thunderstorms of 2017, which are simply issued from quicklooks (http://saetta-lma.aero.obs-mip.fr/; all 2017 data not yet processed). The graph exhibits 2 maxima in June and in September, and a minimum in August (winter is disregarded). From May to August the number of lightning days seems to follow the elevation of the Sun in the sky, i. e. convection is controlled by the flow of solar energy. While from September, the number of lightning days increases significantly on the domain. The corresponding events can either be numerous small storms over the sea or over the relief (Barthlott et al., 2016), or high precipitation events such as those occurring in southern France or in Corsica (Ducrocq et al., 2008; Lambert and Argence, 2008; Nuissier et al., 2011; Ducrocq et al., 2013; Scheffknecht et al., 2016), which are formed under the influence of synoptic flows that interact with the relief of Corsica through low level water vapor fluxes (Adler et al., 2016). The key ingredients that control their development are orographic forcing, low level convergence, strong moisture fluxes and conditionally unstable flow. Furthermore during the autumn season the sea surface is still warm and constitutes a pool of water vapor and energy meanwhile upper level cold air begins to progress from the north, associated with low geopotential height in the region.

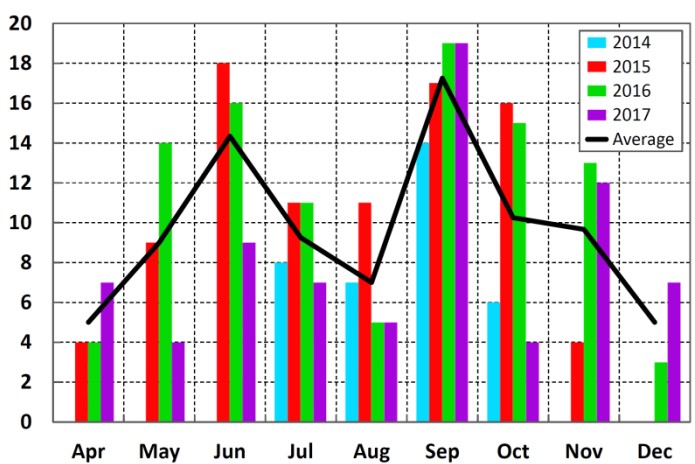

Figure 9: Monthly number of lightning days in the 240 km × 240 km domain of better detection accuracy centered on Corsica for 2014 to 2017 years.

In July the number of lightning days is not the highest (Fig. 9), but since thunderstorms often occur in the same micro-region, the frequency of thunderstorms per pixel in this zone is the highest (Fig. 8). If we add the fact that the number of lightning days in July is particularly homogeneous from year to year (between 7 and 11 over 4 years in Fig. 9), it seems that

this area is the seat of very specific and reproducible processes that lead to this maximum frequency of thunderstorm days in this location in July between 11:00 UT and 14:00 UT.

So to answer the second point of the question asked before (i.e. why thunderstorms are together very active and often present in the north center of Corsica?), one has to analyze the processes involved in convection in July between 11:00 UT and 14:00 UT in the center of the northern part of Corsica. This work is currently being carried out by means of high resolution numerical simulations and will be the subject of a forthcoming paper. The first results (Tidiga et al., 2018) show that the convection follows the setting up of sea breezes which are then channeled through the 3 main valleys, which leads to a strong convergence of low-level moisture flows in the convection trigger zone. This result is to be compared with the influence of valley and slope breezes as suggested by Barthlott and Kirshbaum (2013).

## 4 Specific events report: high altitude discharges

We report hereafter specific events that were detected by SAETTA in 2015. They concern high altitude discharges in the convective core and in the trailing stratiform region of several thunderstorms. Such events raise open questions about the conditions for the triggering of cloud discharges and for thunderstorm electrification.

From 7 to 10 June 2015, Corsica Island underwent 4 consecutive days of intense, stationary, and long lasting diurnal convection over most part of the relief with lightning activity characterized by highest VHF sources between 12 km and 13 km high. The 7 June event lasted 05h50m (from 09:40 UT to 15:30 UT) and exhibited high altitude discharges on top of a convective core and 1 high altitude discharge in the trailing stratiform region (15:26:15 UT). The 8 June event lasted 03h50m from 10:10 UT to 14:00 UT and exhibited high altitude discharges on top of a convective core and 6 high altitude discharges in the stratiform region (12:38:39; 12:43:02; 12:54:05; 12:59:20; 13:21:43; 13:57:18 UT). The June 9 event lasted 06h30m from 09:10 UT to 15:40 UT and exhibited high altitude discharges on top of a convective core and 1 high altitude discharge in the stratiform region (11:58:38 UT). The June 10 event lasted 04h20m from 10:00 UT to 14:20 UT but did not exhibit any specific high altitude discharges.

### 4.1 Event of 8 June 2015: convective surges

Let us focus on 8 June event which presents more specific discharges at high altitude than the other similar events. The environmental wind pattern is constituted of sea and slope breezes on the ground, with a slight wind oriented toward the south appearing around the 700 hPa level, and strengthening as it turns toward the south-west direction at higher levels. The cloud cover and the cumulative VHF sources density are displayed in Fig. 10. The mesoscale convective system is composed of 3 main convective cores embedded in a southward trailing stratiform region. The first convective core #1 appears further south at the beginning of the event and lasts about 50 minutes from about 11:12 UT to 12:03 UT. It is centered on the point

of coordinates $x = 27$ km and $y = 57$ km and exhibits 2 main layers of strong VHF sources densities located at altitudes of about 6 and 10 km. The analysis of the power with which the VHF sources are detected shows that this core and the event as a whole are of regular polarity, i.e. with upper positive charge and lower negative charge associated with those 2 layers of strong VHF sources densities. The 2 other main convective cores centered on points of coordinates $x = -16$ km and $y = 75$ km for the most easterly one #2, and $x = -16$ km and $y = 75$ km for most westerly one #3, start at 11:44 UT and 11:52 UT, respectively, and last until about 13:00 UT by interacting. At the end of the period they are fed on the north side by a line of zonally oriented smaller cells. According to Stolzenburg and Marshall (2008), a mesoscale convective system is "characterized by a leading region of deep convective clouds forming a line or arc, followed by a broad area of deep nimbostratus clouds". Present thunderstorm event matches this conceptual model with deep convective clouds #2 and #3, but exhibits a specific feature due to the presence of a decaying convective cell #1 embedded in the MCS leading region. This feature is though often encountered in typical MCF stratiform cloud of Houze (1993).

| Convective surges | Time period (UT) | Position | | Maximum altitude (km) |
| --- | --- | --- | --- | --- |
| | | x (km) | y (km) | |
| # 1 | 12:00 - 12:03 | -18 | 73 | 13.0 |
| # 2 | 12:09 - 12:13 | -15 | 72 | 12.6 |
| # 3 | 12:16 - 12:20 | -16 | 74 | 12.8 |
| # 4 | 12:26 - 12:28 | -15 | 77 | 12.6 |
| # 5 | 12:33 - 12:37 | -17 | 78 | 14.0 |
| # 6 | 12:40 - 12:43 | -23 | 78 | 12.4 |

Table 2. Serial number, time period, position and maximum altitude of the VHF sources during the 6 convective surges.

The event is characterized by 6 convective surges identified from SAETTA observation (see Table 2). They correspond to high level updraft intensifications on top of a convective core during few minutes, accompanied by upward developing discharges at altitudes that are higher than the whole storm electrical activity (see Krehbiel et al., 2002). All these surges are associated with the northeast main convective core #2 that is located the most windward and that undoubtedly benefits from richer low-level moist fluxes making it more vigorous. These surges have about the same characteristics as the most intense of them (# 5), the VHF sources of which are displayed in Fig. 11 versus altitude (left) and power of detection (right). These sources are located in a small perimeter of approximately 5 km by 5 km (red dots in the left panels of Fig. 11 dedicated to vertical projections) and are present up to 14 km altitude, while the surrounding VHF sources do not exceed 11.5 km altitude. The analysis of the power with which they are detected (Fig. 11 right) shows that the discharges are weakly radiating therefore they likely correspond to positive discharges, i.e. they propagate in a negative charge layer. This

observation is in agreement with the conceptual model of Stolzenburg et al. (1998) and supports their idea that the upper negative charge layer is the typical uppermost charge region in MCS convection. Furthermore, the top left and top right panels in Fig. 11 (i.e. the altitude versus time windows), show that many small discharges appear quasi continuously between 10 km and 13 km altitude during this sequence. Their characteristics are markedly different from typical IC flashes because they are located in a relatively small cloud volume, they seem to trigger at very high altitude and therefore have a very limited vertical and horizontal extension.

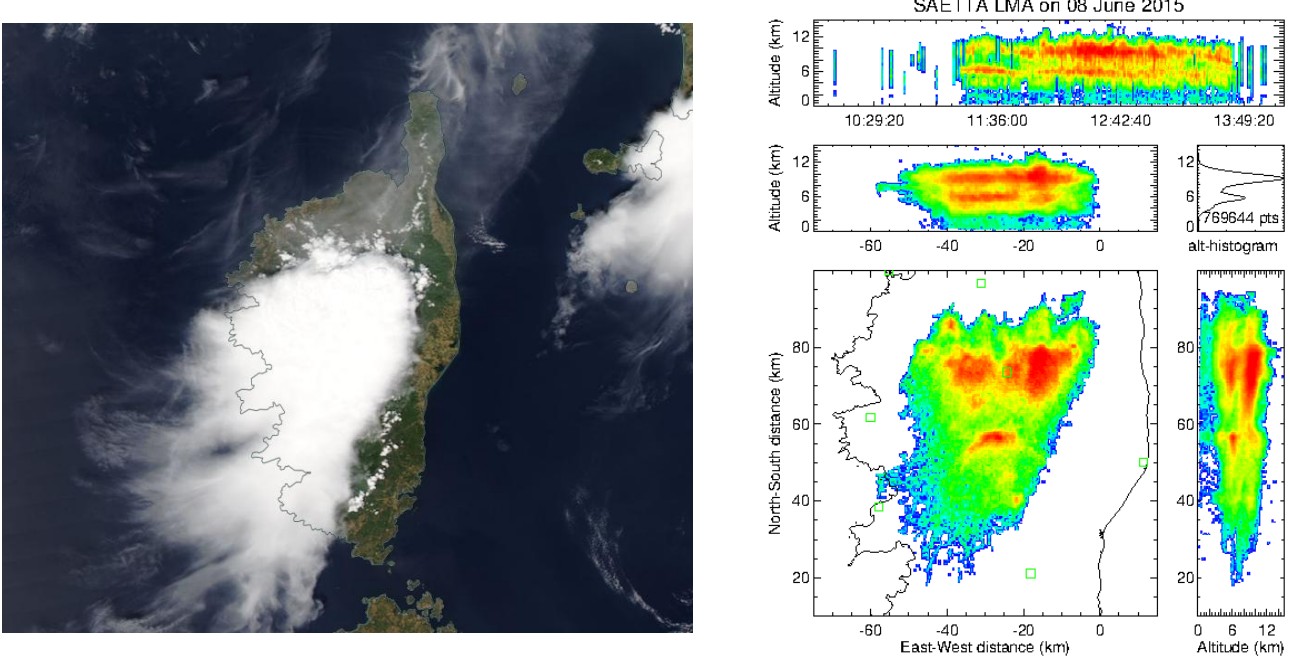

Figure 10: Cloud cover observed around 12:35 UT during the 8 June 2015 storm event by the MODIS/AQUA satellite in the visible wavelength range (NASA Worldview application https://worldview.earthdata.nasa.gov, left), and VHF source density of the event observed by SAETTA (right). Stations location is indicated by green squares

Similar high altitude small discharges were observed by Ushio et al. (2003), MacGorman et al., (2008), Emersic et al. (2011), Calhoun et al. (2012), or MacGorman et al. (2017). These discharges have unusual characteristics, i.e. the corresponding VHF source production is rather continuous, at low rates, with no clear structure, and independent of the flashes at lower altitudes (MacGorman et al., 2017). This behaviour is well illustrated by the quasi continuous small discharges that appear above 10 km altitude in both top panels of Figure 11 (altitude versus time). Associated with so-called rising lightning bubbles (Ushio et al., 2003) or with overshooting tops (e.g. Calhoun et al, 2012; MacGorman et al., 2017), the high VHF source production localized on the top of the convective core is most often associated with a rapid vertical

growth of the storm, i. e. with a convective surge. Calhoun et al (2012) suggested that the decreasing electric threshold for discharge triggering with increasing altitude because of decreasing pressure on one hand (see MacGorman and Rust, 1998), and the charge carried by hydrometeors transported aloft by the updraft and subsequently interacting with the screening layer of charge on the other hand, contributed to this VHF emission production associated to overshooting regions. A comprehensive discussion about the hypotheses concerning charge regions producing such overshooting top discharges is proposed by MacGorman et al. (2017). Anyway, the small size of these discharges is probably due to the small spatial extent of charge regions in the overshooting top.

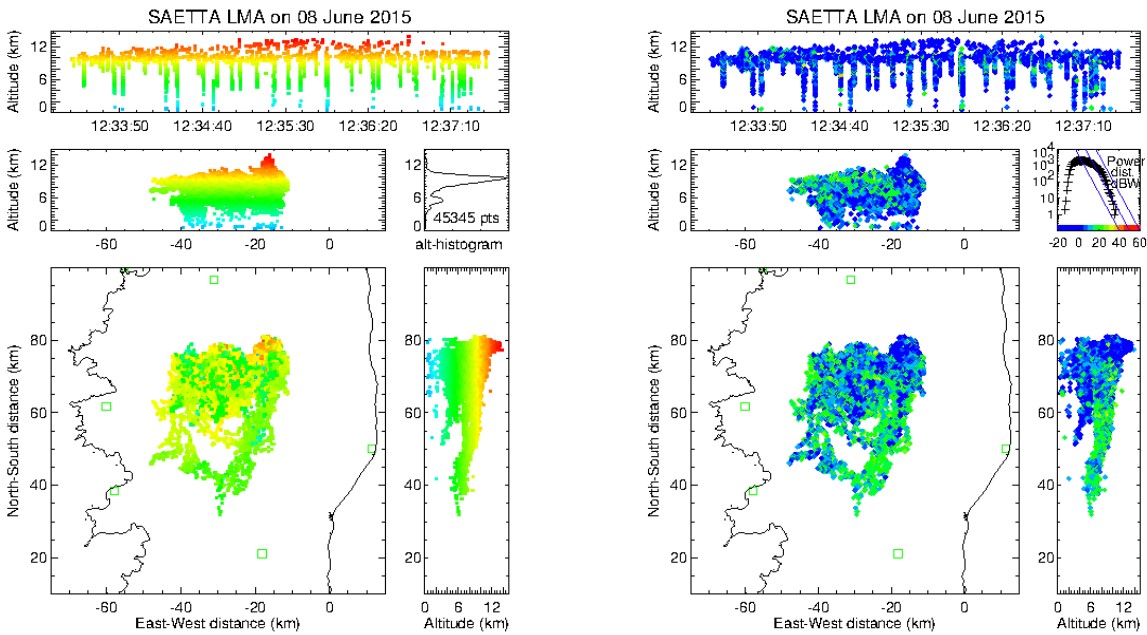

Figure 11: High altitude discharges in the convective zone during the MCS event of 8 June 2015 corresponding to the convective surge #5. VHF sources displayed by dots with color scale corresponding to the altitude of the sources (left), and to the power with which they have been detected (right).

In order to characterize the flashes produced during such convective surges in terms of flash type, triggering altitude, and flash rate, a source-to-flash clustering of lightning was carried out via the XLMA flash algorithm (Thomas et al., 2003). Several algorithms of flash classification have been developed so far (MacGorman et al., 2008, McCaul et al, 2009, Fuchs et al., 2015) but none of them is ideal, especially when the lightning rate is high and the flashes are very close to each other or overlap. The present flash classification does not represent the truth, especially since the classification of small lightning can be affected by the filtering used (minimum number of stations or maximum $\chi^2$). But considered in a qualitative way, it makes

it possible to characterize the types of lightning thus defined according to the evolution of the storm event. The classification criteria are as follows: (i) big flashes (75 or more points); (ii) medium flashes (11-74 points); (iii) small flashes (10 points or less and not isolated); (iv) small-isolated flashes (2-9 points in an active region and duration larger than 1 ms); and (v) short-isolated flashes (2-9 points in an active region and duration smaller than 1 ms). The flash algorithm options (spatial and temporal parameters) are mentioned in Appendix B. The location of the 1st sources is supposed to indicate the location of initialization of the lightning.

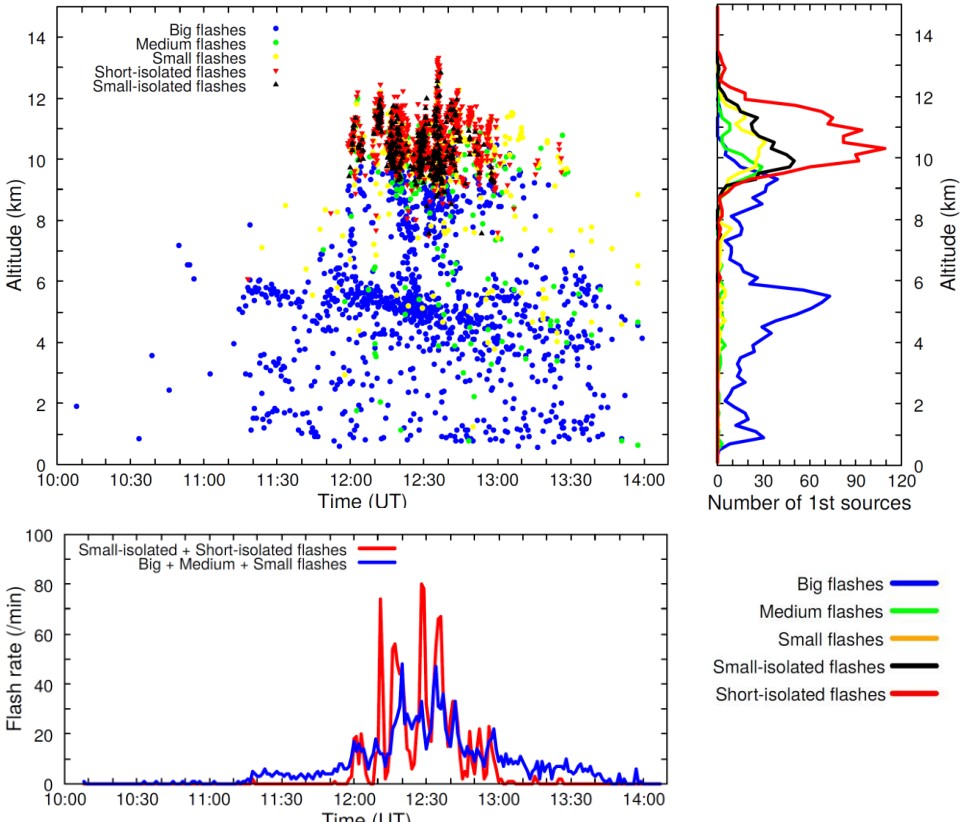

Figure 12: Altitude versus time of the first VHF sources of each flash during the whole 08/06/2015 MCS event (top left), altitude histogram of these first VHF sources with a vertical resolution of 200 m (top right), and 1-minute flash rate versus time (bottom) for Small-isolated and Short-isolated flashes (red) and for Big, Medium and Small flashes (blue). Flash classification: Big flashes (75 or more points); Medium flashes (11-74 points); Small flashes (10 points or less and not isolated); Small-isolated flashes (2-9 points in an active region and duration larger than 1 ms); and Short-isolated flashes (2-9 points in an active region and duration smaller than 1 ms).

The lightning flash rate per minute, calculated for big, medium, and small flashes together is displayed in the bottom panel of Fig. 12 in blue line. It remains rather weak with values lower than 10 min$^{-1}$ until the end of the first main convective core #1 at about 12:00 UT. Then, it exhibits several peaks between 12:00 UT and 13:00 UT with maxima of 48 min$^{-1}$ at 12:20 UT, 33 min-1 at 12:28 UT, 47 min$^{-1}$ at 12:34 UT, and 33 min$^{-1}$ at 12:42 UT. Each of them corresponds to one of the convective surges reported in Table 2. The flash rate corresponding to small-isolated and short-isolated flashes is also displayed in the bottom graph of Fig. 12 in red line for comparison. Its maxima, greater than that of big, medium and small flashes considered together are also in phase with convective surges (see Table 2). The flash rate of small-isolated and short-isolated flashes seems to be a good proxy for the convective surges.

The first sources of flashes have also been extracted via the XLMA flash algorithm in order to identify the preferential altitudes where lightning flashes were triggered. The altitude of those first sources are displayed versus time in the top left graph of Fig. 12 for each kind of lightning flash. Additional information about the number of first sources by flash type is given in the top right graph of Fig. 12 that provides their altitude histogram with a 200 m vertical resolution. The big flashes are triggered from the very beginning of the event until the end (blue dots in top left graph of Fig. 12). During the first convective core #1 between about 11:10 UT and 12:00 UT the altitude of their first sources are distributed over 3 altitude levels (1.5 km, 3 km, and 5.5 km). The 2 lowest levels correspond mainly to Cloud-to-Ground flashes and the highest level to Intra-Cloud flashes. At 12:00 UT big flashes were suddenly triggered at a fourth altitude level laying between 8 km and 10 km high until the end of the more intense convective phase, i. e. 13:40 UT. Note that during the same period from 12:00 UT to 12:40 UT the lowest first sources of big flashes (blue dots) were much less numerous, this corresponds well to the intensification of convection and to a reduction in Cloud-to-Ground flash activity. After 13:40 UT big flashes were again only triggered at 3 altitude levels (1 km, 4 km, and 6 km). As far as medium and small flashes are considered, they were triggered at about all altitudes from 12:00 UT until the end of the event but much less numerous than big flashes, with however maxima at about 9.5 km and 10.7 km, respectively (not clearly visible in top left graph in Figure 12), during the phase of intense convection between 12:00 UT and 13:10 UT. During the same intensive period, numerous small-isolated and short-isolated flashes were triggered between 9 km and 13 km high. The altitude histograms in the top right graph of Fig. 12 show that initiation of big flashes dominates at altitudes lower than 9 km with a maximum around 5500 m altitude. At higher levels corresponding to convective surges activity, big, medium, and small flashes exhibit maxima at about 9300 m, 9700 m, and 10500 m altitude, respectively (the smaller the flash the higher the altitude at which it is triggered), while small-isolated flashes and especially short-isolated flashes exhibit higher maxima at 9900 m and 10300 m altitude, respectively. These results confirm the small scale electrical activity on top of a convective core over small areas centered on convective surge positions indicated in Table 2. Furthermore, it seems that only small-isolated and short-isolated flashes are able to correctly account for this specific high level electrical activity associated with the uppermost part of convective surges.

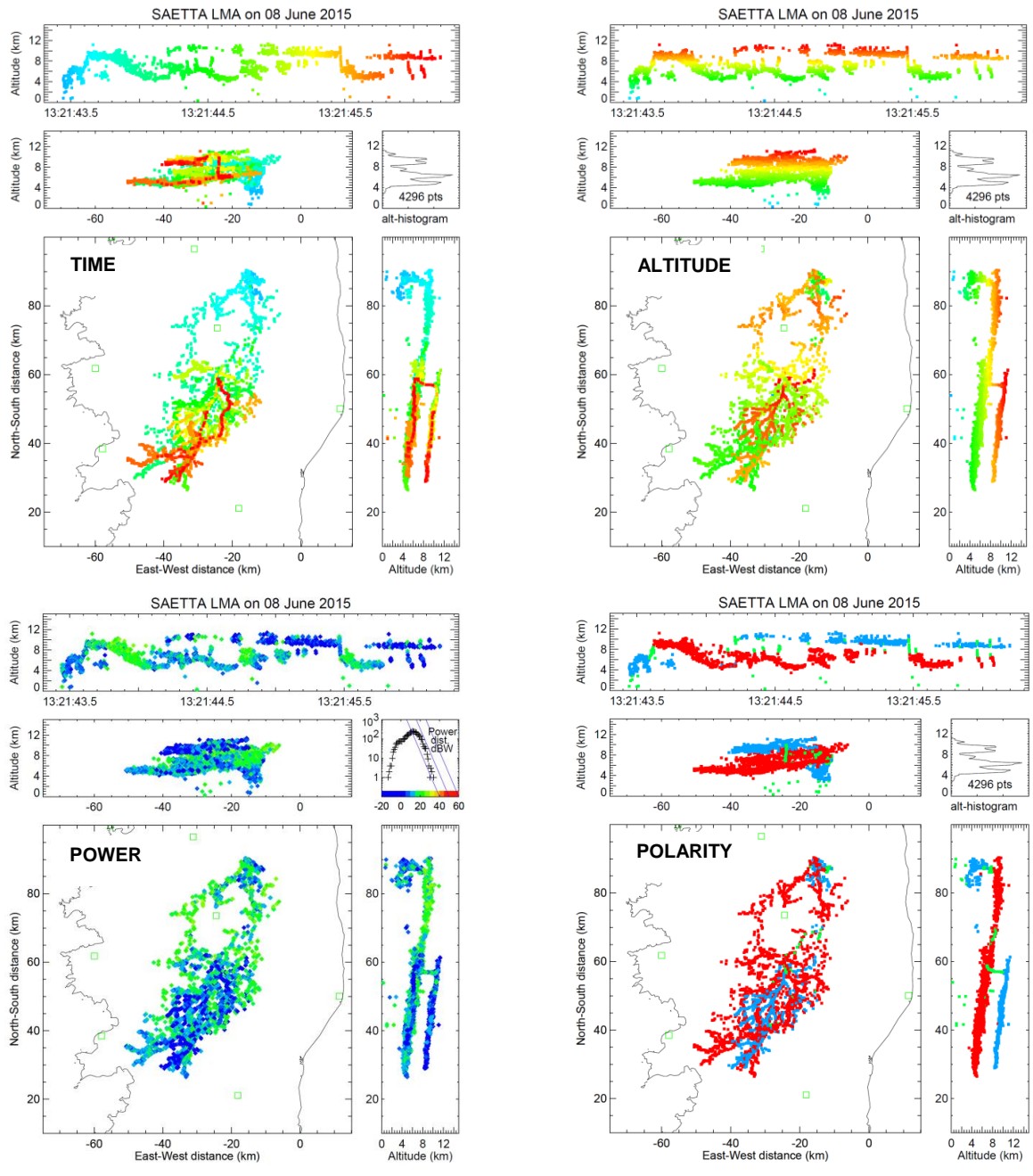

Figure 13: High altitude discharge in the stratiform region during the MCS event of 8 June 2015 at 13:21:43 UT. VHF sources displayed by dots with color scale corresponding to time (top left), altitude (top right), power (bottom left), and cloud charge polarity with positive in red, negative in blue (bottom right).

## 4.2 Event of 8 June 2015: upper level discharges in the trailing stratiform region

During the same event of 8 June 2015, 6 uncommon flashes were detected by SAETTA in the trailing stratiform region of the mesoscale convective system (see Table 3 for detailed temporal and spatial characteristics); 4 of them were triggered during the second half of the phase of intense convection (between 12:00 UT and 13:10 UT, see Section 4.1); and the 2 others were triggered later during the decay phase. Those 6 flashes started like typical IC (or CG for one of them) via a bidirectional leader process (Kasemir, 1950; Israel, 1973; Mazur, 2002; Montanyà et al., 2015) the negative leader of which propagates upward until it reaches the upper positive charge of the regular dipole structure, then propagates away from the convective core #2 into the stratiform region with a low descent associated with the sedimentation of the charged ice particles as described by Ely et al. (2008). However, during the propagation phase through the stratiform region, an upward positive leader - probably issued from a bidirectional leader triggered just above the positive charge layer of the stratiform region - suddenly appears at a distance varying from 14 km (flash #3) to 33 km (flash #2) from the convective core #2 and propagates upward over a vertical distance varying from 3 km (flashes #2 and #3) up to 5 km (flash #5). Then the positive leader spreads in a negative charge layer almost horizontal but with a tilt parallel to that of the positive charge layer below, on altitudes ranging from a maximum of about 12 km to a minimum of about 8 km, and on horizontal distances varying from 10 km (flash #3) to 33 km (flash #5), in several directions but mainly downwind.

Those uncommon flashes can be illustrated by flash #5 that appeared just before 13:21:43.5 UT. The corresponding VHF sources are displayed in Fig. 13 versus time (top left), altitude (top right), power (bottom left), and inferred cloud charge polarity (bottom right). The polarity of the VHF sources (selected by hand with the XLMA software) is deduced from the intensity of the power with which they are detected. Basically most powerful sources correspond to negative leaders that move through positively charged regions while less powerful sources correspond to positive leaders that move through negatively charged regions. More precisely, positive leaders could in fact be the signature of retrograde negative breakdowns located close to the tips of positive leaders. As analyzed by Edens et al. (2012) positive breakdowns do produce weak VHF emissions, however they may be masked by much stronger concurrent VHF emissions from negative breakdowns (see also van der Velde and Montanyà, 2013). But whatever positive leaders are directly or indirectly - via negative retrograde breakdowns at their tips - detected, this possibility does not question their presence and their location. This flash starts as a cloud-to-ground discharge that connects a negative charge layer located between about 4 km and 6 km altitude to the ground around the position defined by $x = -14$ km and $y = 87$ km, and shortly after develops as an intra-cloud discharge with an upward negative leader that rises up to 9 km at 13:21:43.6 where it spreads in a positive charge layer. The propagation of the positive leader into the lower negative charge appears only until 13:21:43.8 while the negative leader travels through the whole stratiform region in a slightly tilted positive charge layer until 13:21:44.7. Before that, a first upward positive leader appears at 13:21:44.2 on top of the positive charge layer at the position defined by $x = -24$ km and $y = 57$ km. Only its upper end can be seen in green in the top panel of the bottom right graph in Figure 13. This upward positive leader reaches the altitude of about 10 km and spreads horizontally. Probably 3 other upward leaders follow the same channel between

13:21:44.3 and 13:21:44.6. Later, 3 downward recoil discharges follow exactly the same path from the uppermost negative charge layer down to the positive charge layer of the stratiform region between about 13:21:45.5 and 13:21:46.1.

Similar flashes were previously reported in the literature but were not analyzed as specific flashes (see Figure 15 in Lang et
al. (2010); Figure 1 in Lang et al. (2011), Figure 7 in Weiss et al. (2012); Figure 12 in Soula et al. (2015); Figure 5 in Lang et al. (2016)). They occurred in different storm organizations in the trailing stratiform region, or in the anvil in case of supercell storm: MCS (Lang et al., 2010; Lang et al., 2011; Soula et al., 2015); supercell (Weiss et al., 2012); or multicell (Lang et al., 2016). The upper level discharge of the flash observed by Weiss et al. (2012) exhibits the typical structure of an intra-cloud flash in an inverted polarity thunderstorm, i.e. an upward propagating positive leader that spreads aloft in a negative charge
layer (in this case over about 25 km) with a shape that looks like an upward water jet. This feature is somewhat different from that of the flash displayed in Figure 13 for which the upward positive leader spreads horizontally aloft over the same distance scale (35 km) but through a thin and very flat layer. The microphysical and dynamical processes must therefore be somewhat different in each of these two cases.

To further identify the discharges processes involved in that complex lightning flash, we performed the time-distance analysis proposed by van der Velde and Montanyà (2013). The calculated horizontal distance between the first source and each other source of the flash is displayed in Fig. 14 versus time as a function of the altitude of the sources (left) and also as a function of the cloud charge polarity in which the leaders propagate (right). In this way, one can easily refer to Fig. 13 (top right and bottom right) to identify the location of each discharge phase. The flash comprises 3 kinds of discharge processes
according to the 3 kinds of slope that can be identified in Fig. 14. Note that positive slopes correspond to discharges propagating away from the first source meanwhile negative slopes correspond to discharge propagating toward the first source. On can first point out 4 main negative leaders (red lines in right graph) propagating away from the first sources in the stratiform region (increasing distance) and toward decreasing altitudes (color from orange to blue in left graph) with a radial speed of about $1.4 \times 10^5$ m s$^{-1}$ (from 0.2 s to 0.75 s for the first one; from 1 s to 1.3 s for the second one; from 1.3 s to 1.45 s
for the third one; and from 2 s to 2.3 s for the fourth one), which is consistent with the measurements by van der Velde et al. (2014). Then, sources associated with the positive leader branch can be identified with a slower increasing distance with time mainly between 0.7 s and 2.7 s in blue color in right graph and in orange and red colors in left graph, i. e. they are localized at high altitude. The corresponding radial speed is about $1.7 \times 10^4$ m s$^{-1}$, which is one order of magnitude less than that of negative leaders. Those positive leaders correspond to the uncommon uppermost altitude discharges in the stratiform region.
At last, one can identify very fast propagating discharges corresponding to the almost vertical lines in Fig. 14 at about 0.8 s; 1.5 s; 2.4 s; 2.5 s; 2.6 s; and between 1.5 s and 1.8 s, the radial speeds of which are about $10^6$ m s$^{-1}$ (between $8 \times 10^5$ m s$^{-1}$ and $1.4 \times 10^6$ m s$^{-1}$). Similar almost vertical lines are visible in Figure 2c and 2d of van der Velde and Montanyà (2013). Those fast propagating discharges may be dart leaders or long recoil events according to van der Velde and Montanyà (2013). The

latter can be distinguished at the end of the flash sequence in the top left graph of Fig. 13: The discharge propagates 3 times consecutively from the south tip of the upper most layer of VHF sources, returns northward to the vertical channel of the initial ascending leader, descends along this channel and propagates southward into the positive charge of the stratiform region (in orange and red). As a matter of fact, the successive passages of these events through the vertical channel can be illustrated by the green dots present in each of the almost vertical lines in the right graph of Fig. 14. All the radial speeds here evaluated are in good agreement with previous observational studies; see van der Velde and Montanyà (2013) for a review.

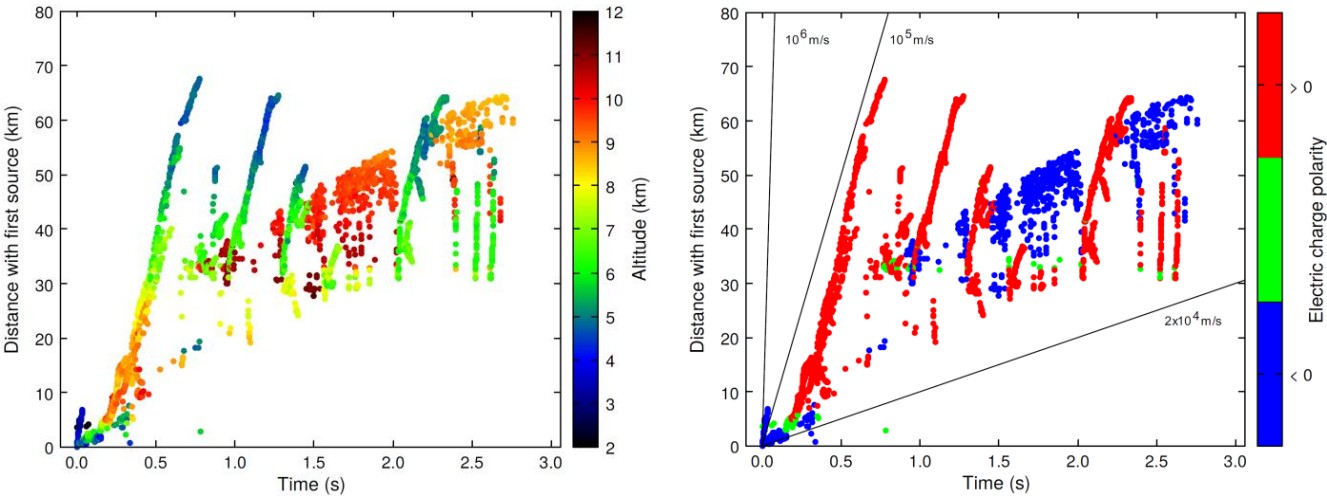

Figure 14: High altitude discharge in the stratiform region during the MCS event of 8 June 2015 at 13:21:43 UT. Horizontal distance between each VHF source and the first source versus time in function of altitude (left) and cloud charge polarity in which the leaders propagate (right). The time is indicated in seconds from the beginning of the flash. The thin black lines indicate slopes corresponding to speeds of $2 \times 10^4$ m s$^{-1}$, $10^5$ m s$^{-1}$, and $10^6$ m s$^{-1}$.

Several questions arise from the observation of such a complex lightning flash involving 2 vertically separated layers of charge in the trailing stratiform region. How a positive leader can propagate upward from the top of the main positive charge layer in that region? What are the mechanisms involved in the formation of the uppermost negative charge layer in that region? Why don't all MCS exhibit such complex lightning flashes? One can refer to the review by Stolzenburg and Marshall (2008) on the charge layers structure in such convective systems, based on in situ balloon-borne measurements. As observed in the present study, lightning in convective systems is most often triggered in or near the convective cores (Ribaud et al., 2016) and can subsequently propagate horizontally into the anvil or into the trailing stratiform region (Carey et al., 2005; Dotzek et al., 2005; Tessendorf et al., 2007; MacGorman et al., 2008, Lang et al 2010, van der Velde et al 2014, Soula et al 2015), following the path of charged ice particles that sediment (Carey et al., 2005; Ely et al., 2008). Kuhlman et al. (2009)

reported observations of in-cloud development of lightning flashes in the anvils of 2 supercell storms. They showed that the convergence of the anvils with opposite polarities of charge at the same altitude could increase electric field magnitude and favor the initiation of distant anvil lightning. But lightning activity in anvils at earlier and latter times was rather supposed to result from some charging mechanism that would be active in anvil as suggested by Dye and Willett (2007), i.e. that anvil

charge would not necessarily originate from transport from the convective core. They also made a comparison with the high electrification of stratiform precipitation regions of mesoscale convective systems where extensive lightning activity develops horizontally (Dotzek et al., 2005, MacGorman et al. 2008) as in the anvils they observed. Recently, Dye and Bansemer (2018, 2019) proposed a conceptual model of mesoscale updraft covering extensive and deep areas during long periods, with non-riming ice collisional charging at mid to upper levels in absence of supercooled liquid water (Luque et al.,

2016), in which larger particles carrying one charge (here positive) fall relative to the smaller particles particle carrying the opposite charge (here negative). Reaching a balance level near the top of the cloud where the updraft has became weak, the small particles with low terminal velocities would accumulate at that level while larger particles with terminal velocities greater than the updraft would sediment downward in the cloud. This scenario would result in the presence of a narrow layer of charge (here negative) near the top of the cloud and a thicker layer of charge (here positive), in a 30 min time period.

| Flash | Flash type | Time (UT) | Duration (s) | First source altitude (km) | Convective core position | | Upper level vertical branch heigh (km) | Vertical branch position | | Distance from vertical branch to convective core (km) | Upper level discharge horizontal extension (km) |
|---|---|---|---|---|---|---|---|---|---|---|---|
| | | | | | $x$ (km) | $y$ (km) | | $x$ (km) | $y$ (km) | | |
| # 1 | IC | 12:38:39 | 1.8 | 8 | -16 | 74 | 3.5 | -17 | 56 | 18 | 16 |
| # 2 | IC | 12:43:02 | 2.0 | 8 | -41 | 77 | 3 | -28 | 47 | 33 | 18 |
| # 3 | IC | 12:54:05 | 1.7 | 7.5 | -14 | 77 | 4 | -23 | 66 | 14 | 10 |
| # 4 | IC | 12:59:20 | 2.5 | 7 | -43 | 72 | 3 | -43 | 46 | 26 | 22 |
| # 5 | CG/IC | 13:21:43 | 2.8 | 4 | -14 | 87 | 5 | -24 | 57 | 31 | 33 |
| # 6 | IC | 13:57:18 | 1.7 | 5 | -39 | 84 | 3 | -24 | 59 | 29 | 21 |

Table 3. Time and spatial characteristics of the 6 upper level discharge observed by SAETTA in the stratiform region of the 8 June 2015 mesoscale convective system.

Observations of the 8 June 2015 mesoscale convective system show the presence of a narrow layer of negative charge near the top of the cloud and a thicker layer of positive charge about 4 km below, with a gap of about 2 to 3 km between them. These features apparently correspond to the conceptual model of Dye and Bansemer (2018, 2019) described above. The geometrical characteristics of the thin upper layer of negative charge must though be confronted to further analysis. For

example, are (i) the horizontal extension (33 km) of this upper layer, (ii) the height (4 km) of the vertical channel through which the discharge connects the two layers, and (iii) the slope of the upper thin layer, consistent with this model of mesoscale updraft? Could the thin upper layer of negative charge not originate from a screening effect by electrostatic influence from above the cloud (Marshall et al., 1989; Wiens et al., 2005)? All these questions remain open and could benefit from a modeling study with a mesoscale cloud resolving model.

One may also wonder if the first convective core #1 that appeared between 11:12 UT and 12:03 UT about 20 km to the south of the 2 other main convective cores #2 and #3 could have played a part in the subsequent charge structure of the trailing stratiform region in which it was embedded. Actually it was still active when those other cores #2 and #3 developed (11:44 UT and 11:52 UT). They produced their first long range lightning that propagated in their common stratiform region at 12:09:56 UT, i.e. only 7 min after the end of the electrical activity of the first convective core #1, and produced their first uncommon high altitude discharge (flash #1 in Table 3) at 12:38:39, i.e. about 30 min later. Therefore, the interaction of this first convective core #1 with the subsequent common trailing stratiform region of the two other cores #2 and #3 should be further explored. Similarly, one could also consider the interaction between both trailing stratiform regions associated with each of the two main convective cores #2 and #3.

## 5 Conclusion and perspectives

Corsica is a very suitable place to study convection in a mountainous maritime environment and to observe the climate trend of convection in Mediterranean region identified as a climatic hot spot (Giorgi, 2006). In 2014, 12 LMA stations constituting the SAETTA network were deployed there to carry out the monitoring of the total lightning activity at high spatial and temporal resolutions. The network has been operational since the summer of 2014, with winter interruptions in the first two years and then permanent operation since April 2016, with the project of operating on long term to try to observe climatic trends. As far as we know, SAETTA is the first LMA network deployed on such a rough terrain with a range of altitude of about 2000 m.

In order to explore the geometric performance of SAETTA we evaluated its line-of-sight visibility by at least 6 stations considering the mask effect of the relief. This is the first time such an exercise has been done for a network of LMA stations. We found that in the range of about 120 km from the center of Corsica, the minimum altitude above which a VHF source can be detected is less than 2 km on average except in some sectors in the south and south-east of Corsica where this altitude can rise 4 to 5 km beyond about 100 km from the center of the island. This geometric performance logically deteriorates when the 3 highest stations are off for wintering from December to March, with a south-west region very poorly documented beyond 100 km from the center of the island (minimum altitude greater than 6-9 km). We also evaluated its location accuracy by means of the geometric model of Thomas et al. (2004) and we compared the results with those concerning the

STEPS network (Lang et al., 2004; Thomas et al., 2004). The performance seem very similar (and even better for the slant range) albeit the SAETTA network is much more geographically extended for the same number of stations. The contribution of the vertical baseline of this mountainous network compared to a totally flat network shows also that the vertical accuracy is significantly improved for sources beyond 150 km. A more comprehensive assessment of SAETTA's performance could undoubtedly benefit from comparisons with trajectories of airliners that can be detected by the network. We plan to perform this kind of comparison with GPS data of commercial flights.

Combining observations from the years 2014 to 2016, we have elaborated a preliminary climatology of total lightning activity on Corsica, which has never been done so far, in a 240 km × 240 km domain. The number of lightning days per square kilometer is dominated by daytime convection over the relief from June to July, with a local maximum at the north center of the island produced in July between 11:00 UT and 14:00 UT. Tidiga et al. (2018) showed - via a numerical study using the cloud resolving model Meso-NH at high resolution - that in absence of synoptic forcing this local maximum is likely due to the low-level convergence of moist air fluxes originated from sea breezes channeled through three main valleys that converge towards each other at this place. Subsequent studies envisaged should analyze the fine-scale impact of synoptic forcing on this scenario. Additionally, the monthly number of lightning days undergoes two maxima: (i) one in June due to daytime convection in phase with the maximum of solar flux at the summer solstice; (ii), and one in September associated with numerous small storms over the sea or with some high precipitation events. Those last events may be associated with high precipitation and flash floods (Scheffknecht et al, 2016) and are the focus of the HyMeX program (Ducrocq et al., 2014).

The present paper also reports unusual lightning events that occurred in a mesoscale convective system on June 8, 2015. Produced during convective surges, the first type of lightning events consisted of numerous VHF sources concentrated on a small perimeter (5 km by 5 km) and protruding from the top of the cloud located at about 11.5 km, up to 14 km from altitude. They correspond to a quasi continuous activity of positive leaders of very limited vertical and horizontal extension reaching an upper layer of negative charge, i. e. probably the typical uppermost charge region in MCS structure (Stolzenburg et al., 1998). The implementation of the XLMA flash algorithm of Thomas et al. (2003) shows that most of this lightning activity consisted of small-isolated and especially short-isolated flashes, which are most commonly disregarded in flashes classification. The second type of lightning events concerns uncommon high altitude discharges in the trailing stratiform region of the mesoscale convective system. A focused analysis is made on one of them that started as a cloud-to-ground flash, propagated upward with a negative leader reaching the upper positive charge of the cloud, then propagated away from the convective core #2 into the stratiform region with a modest descent. At this time a positive leader rose vertically over 5 km and 31 km away of the convective core #2 to reach an uppermost thin layer of likely negative charge, with multiple subsequent recoil phases between this uppermost charge layer and the lower main positive charge layer. Such a complex flash has seldom been observed and published before (best examples in Weiss et al. (2012) and in Lang et al. (2016)) and

was almost never analyzed as a specific flash type that provides information on the upper positive charge layer in the stratiform zone of a MCS. Interestingly, this type of flash may confirm the recent conceptual model of Dye and Bansemer (2018, 2019) that explain such upper level layer of charge in the stratiform region by the action of a non-riming ice collisional charging in a mesoscale updraft. A more detailed assessment of the multiple simultaneaous nearby cells is

required in order to confirm this cause.

SAETTA is now a high performance lightning 3D imager that can serve as a reference for electrical schemes of Meso-NH cloud resolving model (Barthe et al., 2012; Pinty et al., 2013), for operational LF/VLF lightning location systems in this region, for measurement campaigns such as EXAEDRE (https://www.hymex.org/exaedre/) that took place in Corsica in

September and October 2018, and also for the calibration/validation phases of the future observations performed by the optical lightning imager LI on the Meteosat Third Generation geostationary platform (Eumetsat) that will be launched in the forthcoming years.

**Appendix A : Geometric capability of VHF source detection by SAETTA**

In this appendix we describe the geometric method used to estimate the minimum altitude at which a VHF source can be

detected by the SAETTA network in a 240 km × 240 km square area centered on Corsica, with a 5 km horizontal resolution (48 × 48 pixels). All altitudes are here considered above mean sea level (AMSL) and the Earth radius $R_E$ is assumed uniform on the domain. For simplicity, the atmospheric refraction is not taken into account. Therefore the here calculated altitudes overestimate the real minimum altitudes of VHF source detection since electromagnetic waves propagating in the clear sky are actually deflected downwards because of the refractive index gradient that is most often downward directed. The location

of SAETTA stations is provided by their GPS positioning system; their altitude is derived from the data set of the National Institute of Geographical and Forest Information (IGN) via the Geoportail website (https://www.geoportail.gouv.fr/). All geographic positions are converted to Cartesian coordinates using Lambert's conformal conic projection for France (Duquenne et al., 2005). The relief of Corsica is obtained by interpolation – by the Cressmann method with a horizontal resolution of 100 m – of the SRTM digital elevation data (SRTM 90m Digital Elevation Database v4.1) of the Consortium

for Spatial Information CGIAR-CSI (http://www.cgiar-csi.org/data).

The algorithm implemented for this calculation is as follows (see Fig. A1 to identify the different variables named hereafter): (i) a point Px at an altitude $z_{Px}$ above the center of a given pixel is considered; (ii) one looks at whether the direct line of sight between this point Px and each of the SAETTA stations (point St of altitude $z_{St}$) intersects the terrain of altitude $z_r$ or not (for

that the altitude $z_P$ of each point P distributed every 1 km along each line of sight is compared to $z_r$); (iii) if the considered point is visible by less than 6 SAETTA stations, its altitude is increased (increment of 500 m); (iv) the sequence (iii) is

repeated as long as the considered point is visible by less than 6 stations; (v) when the considered point is visible by at least 6 stations, its altitude $z_{Px}$ is the solution for the given pixel and one moves to another pixel.

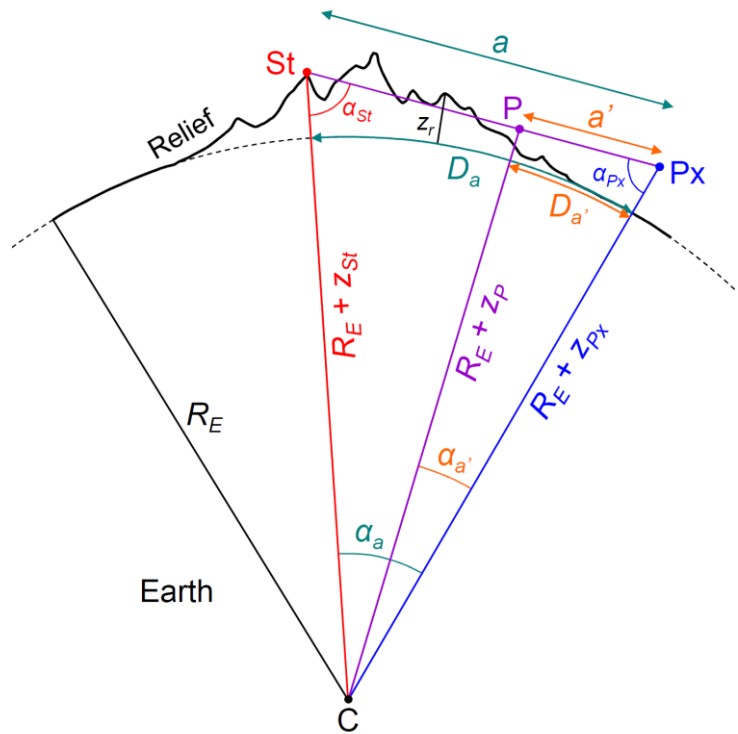

Figure A1: Geometry used in the calculations carried out to evaluate the geometric potential of VHF sources detection by the SAETTA network

The main difficulty that arises in this calculation is to determine the altitude $z_P$ of each point P along the direct line of sight (Px, St) by taking account the roundness of the Earth, the center of which is the point C. Here is how the calculation was conducted. According to Fig. A1, $D_a$ stands for the known geographic distance between the pixel and the station, and $D_{a'}$ stands for the known geographic distance between the pixel and the point P along the line of sight. Assuming the Earth curvature is uniform on the domain, one can deduce the corresponding angles $\alpha_a$ and $\alpha_{a'}$:

$$\alpha_a = \frac{D_a}{R_E} \quad \text{and} \quad \alpha'_a = \frac{D'_a}{R_E} \quad \text{(A1)}$$

Then, the distance $a$ between the points St and Px is deduced from the generalized Pythagorean theorem applied to the triangle (C, St, Px) according to the following expression.

$$a^2 = \left(R_E + z_{St}\right)^2 + \left(R_E + z_{Px}\right)^2 - 2\left(R_E + z_{St}\right)\left(R_E + z_{Px}\right)\cos\alpha_a \quad \text{(A2)}$$

Using the law of sines in that triangle (C, St, Px):

$$\frac{R_E + z_{St}}{\sin\alpha_{Px}} = \frac{R_E + z_{Px}}{\sin\alpha_{St}} = \frac{a}{\sin\alpha_a} \quad \text{(A3)}$$

10  we can deduce the angles $\alpha_{St}$ and $\alpha_{Px}$:

$$\sin\alpha_{St} = \sin\alpha_a \times \frac{R_E + z_{Px}}{a} \quad \text{and} \quad \sin\alpha_{Px} = \sin\alpha_a \times \frac{R_E + z_{St}}{a} \quad \text{(A4)}$$

Using the law of sines in the triangle (C, St, P) gives:

$$\frac{R_E + z_P}{\sin\alpha_{St}} = \frac{a - a'}{\sin\left(\alpha_a - \alpha'_a\right)} \quad \text{(A5)}$$

Using the law of sines in the triangle (C, P, Px) gives:

$$\frac{R_E + z_P}{\sin\alpha_{Px}} = \frac{a'}{\sin\alpha'_a} \quad \text{(A6)}$$

By isolating $a'$ in equation A6 and returning its expression in equation A5, we obtain the unknown $z_P$ that we are looking for:

$$z_P = \frac{a}{\dfrac{\sin\left(\alpha_a - \alpha'_a\right)}{\sin\alpha_{St}} + \dfrac{\sin\alpha'_a}{\sin\alpha_{Px}}} - R_E \quad \text{(A7)}$$

Fig. A2 illustrates the determination of the minimum altitude at which a considered point (here the southwest corner of the domain corresponding to the zero abscissa) can be seen from one SAETTA station (here the Aleria station located on the east coast). The red lines correspond to lines of sight that intersect the relief meanwhile the blue line corresponds to the first altitude (here 12500 m) at which the considered point can be seen by the SAETTA station. The curvature of the lines

5     illustrates the influence of the roundness of the Earth. This approach was used to produce the results shown in Figure A2.

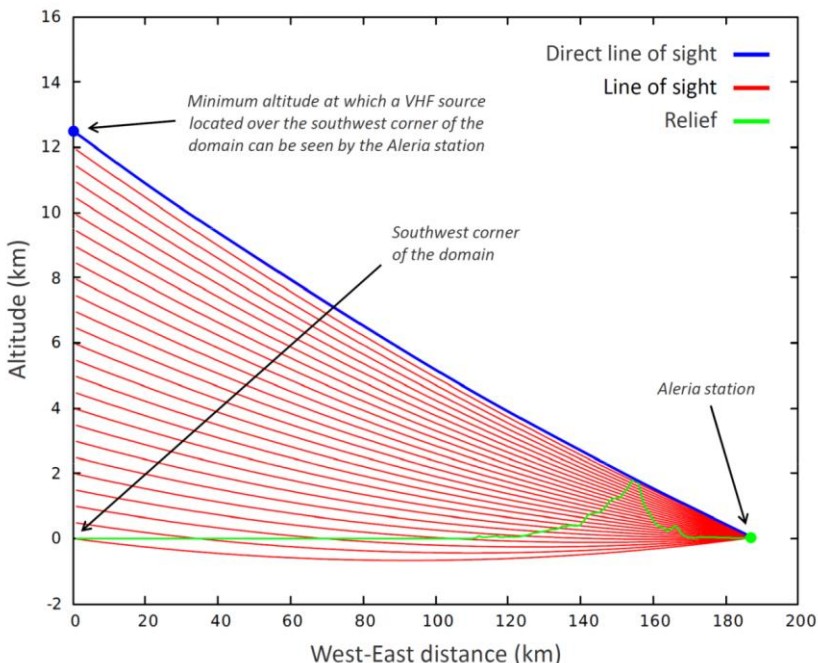

Figure A2: Illustration of the search algorithm for the minimum altitude of direct vision between a pixel and a station (e.g. here between the southwest corner of the domain and the Aleria station on the east coast) taking into account the earth's

10                                                   roundness.

**Appendix B : XLMA Flash Algorithm Options**

| | |
|---|---|
| Max spatial separation for points in a flash (m) | 3000.0 |
| Max altitude separation for points in a flash (m) | 5000.0 |
| Max time separation for points in a flash (s) | 0.15 |
| Max allowable flash length (s) | 3.0 |
| Min acceptable vertical velocity (m/s) | 20000.0 |
| Max number of points in a flash group | 50000.0 |

| | |
|---|---|
| Lat Lon pixel size for density ratios and size | 0.01 |
| Max number of points to be a small flash | 10 |
| Point division between medium and big flashes | 75 |
| Max number of points in a flash fragment or noise | 3 |
| Ratio of number of points between parent and fragment | 15 |
| Max difference in azimuth between parent and fragment (rad) | 0.05 |
| Fraction of spatial separation for sparce connection | 0.50 |
| Min number of points in a sparce connection | 3 |
| Max score for noise | 2 |
| Normal IC altitude (m) divider (75% must be greater than) | 5500.0 |
| Low flash altitude (m) divider (75% must be less than) | 7000.0 |
| Rejoin flashes? | yes |

**Acknowledgments**

Acknowledgements are addressed to SAETTA main sponsors: Collectivité Territoriale de Corse through the Fonds Européen de Développement Régional of the European Operational Program 2007-2013 and the Contrat de Plan Etat Région that funded the CORSiCA project; Collectivité de Corse through the CORSiCA 2017-2019 project; CNRS-INSU through the
HyMeX/MISTRALS program; ANR IODA-MED; CNES through the SOLID project; UPS/Observatoire Midi-Pyrénées; and Laboratoire d'Aérologie. We also strongly thank the many individuals and regional institutions in Corsica (including the Conservatoire du Littoral, Qualitair Corse, and INRA San Giuliano) who host the 12 stations of the network, who helped us to find sites, or who bring us assistance for the logistics during missions in the field. We acknowledge the use of imagery from the NASA Worldview application (https://worldview.earthdata.nasa.gov), part of the NASA Earth Observing System
Data and Information System (EOSDIS). We thank both reviewers for their constructive comments and suggestions that helped us to improve the manuscript.

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
