# Peer review of "SAETTA: high resolution 3D mapping of the total lightning activity in the Mediterranean basin over Corsica, with a focus on a MCS event"

_Atmospheric Measurement Techniques, 2019_

## Referee Comment (RC1) · Anonymous Referee #1 · 8 Jul 2019

General comments:

This paper introduces the SAETTA 3D Lightning Mapping Array installed on the island of Corsica and discusses its general performance and climatology, and also illustrates the data by two case studies. It is found that in summer afternoons, a thunderstorm often forms at the particular location on the island, where the intersection of 3 wide valleys brings together the moist air by sea breeze circulation. One case study showcases the use of the rates of lightning flashes separated by size and finds surges of rates of tiny flashes confined to high altitudes during intense convective development. The other case study reveals a negative upper charge layer in the stratiform region of a

[Figure]

Mesoscale Convective System (MCS) by activity of positive or recoil leaders detected by the LMA. In general, the paper is a good contribution. The selected case studies are interesting, but some earlier papers on the matter of small discharges as well as some with examples of negative charge layers in the stratiform region are missing. Also, the inclusion of radar images could give the case studies some more depth. The text also needs a number of technical corrections, suggested below.

Specific comments:

1. Aircraft-related trails claimed in statistical results, but this has not been detailed with an example.

2. Surges of tiny flashes. Other works have described them before, it is possible to add references and some discussion.

3. Stratiform high-altitude positive leaders. Other works have shown these before. Is it possible to investigate if they spawn off the first negative leader branch and then grow upward and sideward? This mechanism looks similar to the case of negative leader suddenly spawning (up or down) from a horizontal positive leader branch as in examples in van der Velde and Montanyà (2013).

Radar analysis could help provide some context for topics 2 and 3, in terms of confirming cloud top altitude and evolution stage. Time-distance plots need the slopes indicated to have a proper reference. The text may be written a bit more concisely (remove some details or similar paragraphs).

Technical corrections:

12: 80 us is the resolution per station, but after locating the sources, this is not the resulting temporal resolution, which is decreased. 16: masking (also Page 5, line 10) 18: integrated across the domain; reaches a first maximum 19: intersection of 21: lightning 22: and includes some high precipitation events

5: relaxation but perhaps a limiter? 14-15: thunderstorm cell; remove On another hand 22: high precipitation; climate change 24-25: network instead of instrument 27: which means 31: put comma after summits; meters altitude

8: carrying humid air 14+18: original could be unique or unusual

4: lightnings → lightning processes

Table 1: "Gap" → vertical difference. But the table is not very interesting, the two columns are almost identical. Is it necessary?

13: average minimum altitudes

6: at best → in the optimal situation 8: Koshak (also later on) 13: is therefore a suitable tool 18: north, south, east, west are not to be capitalized (also occurs later) 20: remove of before altitude

Figure 4: maybe provide a figure with the difference between the two? At first sight they look almost identical. Also: Thomas et al. 2004 (also on Fig 3) 19: better goodness of fit, lower reduced chi-squared

4: From here on, 14: this seems to be the first sample. 15: Looking at the top frame

3: check if this is the continuation of the same leader as before using a time-distance plot (van der Velde and Montanya 2013)

2: The sample size is not yet large enough to call this climatology 9: north, forming. . . 10: Need to show more detail and example of aircraft. It seems unlikely to result in more sources when a lightning flash produces far more. Could be interference from local noise that creates patterns?

Page 14 10: stormy day. . . maybe better lightning day or thunderstorm day?

2: southern France or in the south of France. 5-6: These ingredients are universally indicated in textbooks. The reference provided is not the real source. 7: begins

8-12: duration 3h50m 10: on top of a convective core 17: northerly wind 18: turns to northeasterly 19: is composed of general: it would help to have radar images and number the cores discussed in the text, then coordinates are not needed. How are convective surges defined? Just from LMA data? 28: Marshall (also elsewhere) 29: thunderstorm structures (charge structures?)

14: source density In this section or discussion, make references to small discharge surges, like Ushio et al 2003, Emersic et al 2011, MacGorman et al 2017

Fig 11 right, power: what do the diagonal lines mean here?

m-1, min-1... make superscript 8: are in phase with convective surges. ... But how can we confirm this? Radar?

14: big flashes triggering dominates – hard to read, do the authors mean initiation of big flashes? 15-18: too much detail and everything looks similar. Needed? 20: convective surge positions

1-2: over a vertical distance varying

6: in this case probably positive leaders themselves because of their very low power. 7: cloud-to-ground 9: intracloud 11: slightly tilted 12: does the positive leader really grow upward from the negative leader altitude? The graph suggests not (unclear). The vertical channel is mostly a later downward negative leader. On top of. 23: the slopes are missing in Fig 14 27: rather fast. ... This is still quite normal speed. See also van der Velde et al. 2014 27: 1.4x105 use superscript 28: Then, sources associated with the positive leader branch... 35: superscript

4: descends along... and propagates... into 6: nice to see that! It could be interesting to select this point as the origin of an alternative time-distance plot. See also van der Velde et al (2014)

2: needs rewriting, as one answer is also the question 7: van der Velde et al 2014, Lang et al 2010, Lu et al 2013, Soula et al 2015 could also fit 29: Could ... not ? 33: one may wonder

5: Similarly/Additionally . . .

1: geometric performance 5: This performance 16: 2014-2016; a preliminary climatology

5: in van der Velde and Montanya 2013 a negative leader develops vertically from a positive leader, it seems here we have the positive equivalent. Interesting. 7-8: see provided references with similar discharges

References

Ushio, T., S. J. Heckman, H. J. Christian, and Z. I. Kawasaki, 2003: Vertical development of lightning activity observed by the LDAR system: Lightning bubbles. J. Appl. Meteor., 42, 165–174.

Emersic, C., P. L. Heinselman, D. R. MacGorman, and E. C. Bruning (2011), Lightning activity in a hail‐producing storm observed with phased‐array radar, Mon. Weather Rev., 139, 1809–1825, doi:10.1175/2010MWR3574.1.

MacGorman, D. R., Elliott, M. S., and DiGangi, E. ( 2017), Electrical discharges in the overshooting tops of thunderstorms, J. Geophys. Res. Atmos., 122, 2929– 2957, doi:10.1002/2016JD025933.

Soula, S., Defer, E., Füllekrug, M., Velde, O., Montanya, J., Bousquet, O., Mlynarczyk, J., Coquillat, S., Pinty, J.‐P., Rison, W., et al. ( 2015), Time and space correlation between sprites and their parent lightning flashes for a thunderstorm observed during the HyMeX campaign, J. Geophys. Res. Atmos., 120, 11,552– 11,574, doi:10.1002/2015JD023894.

Lang, T. J., W. A. Lyons, S. A. Rutledge, J. D. Meyer, D. R. MacGorman, and S. A. Cummer (2010), Transient luminous events above two mesoscale convective systems: Storm structure and evolution, J. Geophys. Res., 115, A00E22, doi:10.1029/2009JA014500.

Lang, T. J., J. Li, W. A. Lyons, S. A. Cummer, S. A. Rutledge, and D. R. MacGorman (2011), Transient luminous events above two mesoscale convective systems: Charge moment change analysis, J. Geophys. Res., 116, A10306, doi:10.1029/2011JA016758.

Lang, T. J., Lyons, W. A., Cummer, S. A., Fuchs, B. R., Dolan, B., Rutledge, S. A., Krehbiel, P., Rison, W., Stanley, M., and Ashcraft, T. ( 2016), Observations of two sprite‐producing storms in Colorado, J. Geophys. Res. Atmos., 121, 9675– 9695, doi:10.1002/2016JD025299.

van der Velde, O. A., J. Montanyà, S. Soula, N. Pineda, and J. Mlynarczyk (2014), Bidirectional leader development in sprite‐producing positive cloud‐to‐ground flashes: Origins and characteristics of positive and negative leaders, J. Geophys. Res. Atmos., 119, 12,755–12,779, doi:10.1002/2013JD021291.

---

## Referee Comment (RC2) · Anonymous Referee #2 · 18 Jul 2019

Manuscript amt-2019-192

Title: SAETTA: high resolution 3D mapping of the total lightning activity in the Mediterranean basin over Corsica, with a focus on a MCS event

Authors: Sylvain Coquillat, Eric Defer, Pierre de Guibert, Dominique Lambert, Jean-Pierre Pinty, Véronique Pont, Serge Prieur, Ronald J. Thomas, Paul R. Krehbiel, William Rison

The material presented in this manuscript provides (1) a detailed technical background and performance summary for the SAETTA LMA network; (2) a climatological summary of total lightning observations for its early years of operation in its mixed moun-

tainous/maritime operating region; and (3) a case study of an MCS that occurred in 2016 and exhibited interesting/unique flash-level behavior. This is a wonderful collection of materials that will likely make this paper the essential background reference for future users of this 3-dimenstional total lightning mapping network. The tri-level flashes discussed in Section 4.2 are unique (in the eyes of this reviewer). This material clearly deserves to be published, and is probably already being referenced in manuscripts in-preparation.

When reading this manuscript, I struggled with two competing reactions: the breadth and depth of material that made it a bit "heavy" to this reader, which was at odds with several of my annotations that implied additional work and refinement of presentation. I am sure that the authors had similar struggles when preparing the manuscript. The review comments that follow do not reconcile this issue. Sorry. However, I leave these comments in the good hand of the authors to pick-through and decide what works for them. I do not need to review or accept the changes that they make.

Broad Issues:

1. The "tri-level" (stacked?) flashes that were found to develop and then propagate in the trailing stratiform region may not be well-explained by conditions leading to the ice-based charging mechanism suggested by Dye and Bansemer (2019). The authors' discussion between line 25 on page 25 and line 7 on the following page describe other factors that require more-detailed analyses before interactions between separate-but-overlapping storms and upper-level screening can be excluded. The vertical separation of the two upper charge regions associated with the upper-level flashes (I think that I see about 4 km in Figure 13 – not the "2-3 km" that they indicate on Page 25, line 22) seems rather large for screening, and rather and high for charge separation by mesoscale updrafts. Overall – my only suggestion is that the authors "soften" the statements in the abstract (lines 26-27 in page 1) and conclusions (lines 8-10 on page 25), instead of the current rather-direct attribution to this effect.

2. The description and performance characterization of the LMA network in Section 2 is excellent, but I have two issues that are worth mentioning. First, on lines 1-2 on page 4, the authors state a single benefit of having 12 stations rather than fewer stations. There are other good reasons worth mentioning, such as (1) redundancy/reliability (short-term and long-term failure), (2) the effect of localized high-rate storms on a sensor's contribution to more-distant activity, and (3) the improved geometry for geo-location of distant lighting while maintaining height accuracy for nearby low-altitude lightning channels. I am sure that the authors can think of other benefits. The second issue relates to the depiction of vertical accuracy only at 10 km altitude. The vertical accuracy will be worse at about 3 km height (MSL) – above all the mountains. Users of these data would benefit from understanding this issue, either through additional figures or (at least) some words by the authors.

3. It would be nice if the first use of the xlma plots (Figure 5 for density and figure 6 for sources) were described in a bit more detail, including the distance (rather than lat-lon) scales, and then attributed to Ron Thomas and his xlma program.

4. Most of the figures will be difficult for the reader to interpret. The fonts are too small to be read; there is wasted space (large separation between panels) that should be filled with real content (e.g., Figures 7 and 8, among others) ; It is difficult to see the overshooting-tops in Figure 10; the color separation in Figure 9 makes it difficult to see the different years; and the "full xlma figures" maintain a lot of content (wasted visual space) that is not central to that points discussed by the authors (Figures 11 and 13). Maybe all figures should be reviewed by an author or colleague that never got the chance to manually zoom-in on the key features, so that they have the same disadvantages as the future readers.

5. The terrain blockage analysis is an important element of this work. The technical discussion in the appendix does not mention how the authors handled refraction at VHF. Was it ignored, or was the radius of the earth adjusted (increased) to provide a simple correction for this? It would be interesting to know if sources were actually

located at lower-than-expected heights.

Lesser Issues:

6. The end date for the climatology (2016) seems odd, given the availability of data for 2017 and 2018. A sentence providing a rationale would be helpful. It might help explain the awkward statement on line 15 of page 15 ("easy to extract") regarding the 2017 storm-day data.

7. There are several papers in the reference list that are not cited in the paper

8. I have a number of lesser comments, minor corrections, and editorial suggestions in a hand-annotated version of the manuscript. So that this review can be timely, a PDF scanned copy of this annotated version will be provided separately.
* * *

---

## Referee Comment (RC3) · Anonymous Referee #2 · 18 Jul 2019

The comment was uploaded in the form of a supplement: https://www.atmos-meas-tech-discuss.net/amt-2019-192/amt-2019-192-RC3-supplement.zip
* * *

---

## Author Comment (AC1) · 16 Sep 2019

Sunday, 15 September 2019

ATMOSPHERIC MEASUREMENT TECHNIQUES DISCUSSIONS

Manuscript amt-2019-192

Title: "SAETTA: high resolution 3D mapping of the total lightning activity in the Mediterranean basin over Corsica, with a focus on a MCS event"

Authors: Sylvain Coquillat, Eric Defer, Pierre de Guibert, Dominique Lambert, Jean-Pierre Pinty, Véronique Pont, Serge Prieur, Ronald J. Thomas, Paul R. Krehbiel,

[Figure]

William Rison

Dear Associate Editor,

We are very grateful to Referee #1 for his criticisms and suggestions that we tried to take into account to improve the manuscript significantly. We also thank Referee #1 for the numerous corrections and references he proposes. You will find below our item-by-item response (indicated by *) to the comments and recommendations of Referee #1. The proposed modifications appear in blue font in the revised manuscript in order to readily identify them.

General comments:

This paper introduces the SAETTA 3D Lightning Mapping Array installed on the island of Corsica and discusses its general performance and climatology, and also illustrates the data by two case studies. It is found that in summer afternoons, a thunderstorm often forms at the particular location on the island, where the intersection of 3 wide valleys brings together the moist air by sea breeze circulation. One case study showcases the use of the rates of lightning flashes separated by size and finds surges of rates of tiny flashes confined to high altitudes during intense convective development. The other case study reveals a negative upper charge layer in the stratiform region of a Mesoscale Convective System (MCS) by activity of positive or recoil leaders detected by the LMA. In general, the paper is a good contribution. The selected case studies are interesting, but some earlier papers on the matter of small discharges as well as some with examples of negative charge layers in the stratiform region are missing. Also, the inclusion of radar images could give the case studies some more depth. The text also needs a number of technical corrections, suggested below. * We thank referee #1 for informing us about previous observation that are missing in the paper and for which he kindly gave the references. We carefully explored the literature and modified the manuscript accordingly. * We agree with Referee #1 about the interest of radar images but unfortunately Corsica did not have good radar coverage from the Météo France

radar network in 2015 (see Specific comment #3 below). We added a satellite image that provides a good idea of the 8 June 2015 event studied in present paper.

Specific comments:

1. Aircraft-related trails claimed in statistical results, but this has not been detailed with an example. * We deleted the sentences relative to the aircraft trajectories in order to avoid the addition of new figures since the paper is already long. In fact this point does not really matter in this article. It will matter later if we can get the flight data to evaluate the location efficiency of SAETTA. For information, the flight we were talking about produced an exceptionally huge amount of VHF sources (406210 sources, see figure at the end of present document).

2. Surges of tiny flashes. Other works have described them before, it is possible to add references and some discussion. * As suggested by Referee #1, whom we thank for the references they gave us, we added a paragraph with some discussion about previous observation available in the literature.

3. Stratiform high-altitude positive leaders. Other works have shown these before. Is it possible to investigate if they spawn off the first negative leader branch and then grow upward and sideward? This mechanism looks similar to the case of negative leader suddenly spawning (up or down) from a horizontal positive leader branch as in examples in van der Velde and Montanyà (2013). * We added references to previously observed flashes of the same type in a new paragraph of Section 4.2 and also added a comparison with figures by van der Velde and Montanyà (2013) in the following paragraph. * The analysis of these flashes certainly requires a further detailed study to precisely compare the processes that are involved in present flash and in the events reported by van der Velde and Montanyà (2013). For that, we would need more information about the examples mentioned by Referee #1. We have identified similarities with events reported in their Figures 2c and 2d and Figure 6 (event M). We suppose that Referee #1 focuses on the M (probably also N) event reported by van der Velde

and Montanyà (213). It would probably be interesting to compare the flashes by means of XLMA low speed animations for clearly identify each phase of those complex discharges. We think that this task deserves much more time and could be the subject of a new article in order to go further into the details of lightning physics, maybe through a collaboration with Oscar van der Velde and Joan Montanyà..

Radar analysis could help provide some context for topics 2 and 3, in terms of confirming cloud top altitude and evolution stage. Time-distance plots need the slopes indicated to have a proper reference. The text may be written a bit more concisely (remove some details or similar paragraphs). * In 2015, radar observation in Corsica was only available from the Aleria radar (located at low altitude on the eastern coast of Corsica) and Collobrières radar (located on the South-East coast of France at about 270 km from the center of Corsica). The former is close to the relief that acts as a mask and prevents from scrutinizing the depth of the events located on the West part of the island, the latter is very far from Corsica and does not bring good information. Only fragments of the cells were available from both radars so that we did not display those graphs. In order to bring a glimpse of the cloud formation during the 8 June 2015 MCS event, we displayed in Figure 10 the cloud cover observed around 12:35 UT during the 8 June 2015 storm event by the MODIS/AQUA polar orbiting satellite in the visible wavelength range. Unfortunately, only one passage could observe the event (the passage of MODIS/TERRA was too early). * As suggested by Referee #1 we modified the time-distance plot of Figure 14 right (distance between each VHF source and the first source versus time in function of altitude cloud charge polarity) in order to display reference slopes corresponding to speeds of $2\times10^4$ m s-1, $10^5$ m s-1, and $10^6$ m s-1 (same slopes as in van der Velde and Montanyà, 2013).

Technical corrections:

12: 80 us is the resolution per station, but after locating the sources, this is not the

resulting temporal resolution, which is decreased. 16: masking (also Page 5, line 10) * 12: The time resolution has been corrected. * 16: "Mask" has been replaced by "masking" in the abstract and in page 5.

18: integrated across the domain; reaches a first maximum 19: intersection of 21: lightning 22: and includes some high precipitation events * 18: We replaced "on the whole" by "across the". * 19: We replaced "crossroad" by "intersection". * 21: We replaced "lighting" by "lightning". * 22: We replaced "precipitating" by "precipitation".

5: relaxation but perhaps a limiter? 14-15: thunderstorm cell; remove On another hand 22: high precipitation; climate change 24-25: network instead of instrument 27: which means 31: put comma after summits; meters altitude * 5: We added "and a limiter" after "relaxation". * 14: We replaced "thundercell" by "thunderstorm cell". * 14-15: We deleted "On another hand". * 22: We replaced "highly precipitating" by "high precipitation" and "climatic" by "climate". * 24-25: We replaced "instrument" by "network". * 27: We replaced "that" by "which". * 31: We inserted the coma and deleted "of".

8: carrying humid air 14+18: original could be unique or unusual * 8: We replaced "humidity" by "humid air". * 14+18: We replaced "original" by "unusual".

4: lightnings ! lightning processes * 4: We replaced "lightings" by "lightning". We don't think that the word "processes" is necessary here even though, indeed, each lightning comprises numerous processes that multiply the number of VHF sources emitted.

Table 1: "Gap" ! vertical difference. But the table is not very interesting, the two

columns are almost identical. Is it necessary? * Table 1: We replaced "Gap" by "Vertical difference" in Table 1 and in its caption. We would have liked to readily have this kind of information about the other LMA networks... So yes, we think that it could be interesting for other researchers in case of future comparisons.

13: average minimum altitudes * 13: We replaced "a minimum average altitude" by "average minimum altitudes".

6: at best ! in the optimal situation 8: Koshak (also later on) 13: is therefore a suitable tool 18: north, south, east, west are not to be capitalized (also occurs later) 20: remove of before altitude * 6: We replaced "at best" by "in the optimal situation". * 8: We replaced "Koshack" by "Koshak" and also page 6 line 5 of the initial manuscript. * 13: We replaced "thus a good" by "therefore a suitable". * 18: We changed those capital letters to small letters in the whole manuscript. * 20: We deleted "of" before "altitude.

Figure 4: maybe provide a figure with the difference between the two? At first sight they look almost identical. Also: Thomas et al. 2004 (also on Fig 3) 19: better goodness of fit, lower reduced chi-squared * Figure 4: Indeed both figures are very similar over and close to Corsica but the patterns corresponding to larger errors (> 400 m) a clearly different. We corrected the reference to Thomas et al. (2004). * 19: We corrected the sentence accordingly.

4: From here on, 14: this seems to be the first sample. 15: Looking at the top frame * 4: We replaced "there" by "here on". * 14: This lightning was indeed detected by SAETTA during its first month of operation. Does Referee #1 think we should mention this information? Or do we correctly understand his comment? * 15: We deleted

"When".

3: check if this is the continuation of the same leader as before using a time-distance plot (van der Velde and Montanya 2013) * 3: The considered graph (left graph in Figure 6) is not of importance in the paper. As a matter of fact this graph is only presented to show a typical CG flashes and is not a basis for a fine process analysis. Therefore we didn't add the plot proposed by Referee #1. However, the flash concerned is very similar to the event M reported by van der Velde and Montanya (2013) in their Figure 6. The time-distance plot should thus be similar to the one of the M event in their Figure 7, except that no subsequent CG occurs after the one at the beginning of the flash. We added this reference in the manuscript. To go further into details, one could mention that both present flash and the M event reported by van der Velde and Montanyà (2013) similarly exhibit a sudden multidirectional branching of a negative leader that rises the altitude of the upper positive charge layer in which it spreads. In the present case, the upward leader produces 3 branches, the first of them seems to be characterised by a very high ascent speed (probably faster than 105 m/s) and reaches the highest altitudes, probably subject to more intense electrical conditions. Its path is subsequently used in the end of the flash by a recoil leader. But so many details together with an additional plot may not be absolutely necessary and would lengthen the paper when this one is already very long.

2: The sample size is not yet large enough to call this climatology 9: north, forming... 10: Need to show more detail and example of aircraft. It seems unlikely to result in more sources when a lightning flash produces far more. Could be interference from local noise that creates patterns? * 2: We replaced "It is far from representing a sufficient sample to call this climatology" by "The sample size is not yet large enough to call this climatology". * 9: We replaced "and" between "north" and "forming" by a comma.

* 10: As previously mentioned (specific comment 1 above), we deleted the sentences relative to the aircraft trajectories in order to shorten the paper and to avoid the addition of new figures since the paper is already long. But for answering to reviewer #1, the considered flight surprisingly produced an exceptionally huge amount of VHF sources (406210 sources, see figure at the end of present document). The aircraft probably encountered a large amount of ice crystals during its path through the trailing strati-form zone of the storm and underwent an intense collisional charging that could have produced numerous sparks from its wings. Maybe the detection threshold of SAETTA was also relatively low at that moment.

10: stormy day... maybe better lightning day or thunderstorm day? * 10: We here re-placed "stormy day" by "lightning day", and also in other 12 locations in the manuscript.

2: southern France or in the south of France. 5-6: These ingredients are universally indicated in textbooks. The reference provided is not the real source. 7: begins * 2: We replaced "in South of" by "southern". * 5-6: We agree with Referee #1. This sentence - not clear enough - was just to mention that the storm events studied here are similar to those studied in southern France during the HyMeX program, for which the here listed ingredients are more or less of prime influence according to the atmospheric and orographic conditions. They have been identified so in Ducrocq et al. (2008), especially for the so-called "Cevenol events". We deleted the reference in order to avoid any mistake. * 7: We replaced "begin" by "begins".

8-12: duration 3h50m 10: on top of a convective core 17: northerly wind 18: turns to northeasterly 19: is composed of general: it would help to have radar images and number the cores discussed in the text, then coordinates are not needed. How are

convective surges defined? Just from LMA data? 28: Marshall (also elsewhere) 29: thunderstorm structures (charge structures?) * 8-12: We modified the four durations according to the suggestion of Referee #1. * 10: We replaced "on top of convective core" by "on top of a convective core" 3 times in the same paragraph. * 17-18: Wind directions have been modified according to Referee #1 and Referee #2 suggestions. We chose potentially unambiguous formulations ("wind oriented toward the south"; "toward the south-west direction"). * 19: We replaced "constituted" by "composed". * General: As far as radar images are considered, we agree with Referee #1, see our response to the Specific comment #1 above. As suggested we labelled #1, #2, and #3 the three convective cores everywhere in the manuscript, it is very helpful for the reader. Without radar images, we think it's better to keep the information about the coordinates. The convective surges are now defined at the beginning of the 2nd paragraph in Section 4.1 (just from LMA data). * 28: We replaced "Marshal" by "Marshall" everywhere in the manuscript. * 29: The sentence has been modified according to a suggestion by Referee #2. OK with Referee #1, Stolzenburg and Marshall (2008) addressed the question of electrical structure in their paper.

14: source density In this section or discussion, make references to small discharge surges, like Ushio et al 2003, Emersic et al 2011, MacGorman et al 2017 * 14: We replaced "sources" by "source". * In this section or discussion: We added a new paragraph inside which previous observation available in the literature are now discussed, as suggested by Refere #1.

Fig 11 right, power: what do the diagonal lines mean here? * Fig 11 right: They must be references for the slope of the power distribution tail. We don't comment this kind of information. We can erase those lines if Referee #1 thinks it's necessary.

m-1, min-1... make superscript 8: are in phase with convective surges... But how can we confirm this? Radar? * m-1, min-1...: We made superscripts and corrected the first one (m-1 -> min-1). * 8: We added a reference to Table 2 which provides the periods of occurrence of the surges. Those periods match the maxima of the red curve in the bottom graph of Figure 12.

14: big flashes triggering dominates – hard to read, do the authors mean initiation of big flashes? 15-18: too much detail and everything looks similar. Needed? 20: convective surge positions * 14: We replaced "big flashes triggering dominates" by "initiation of big flashes dominates". * 15-18: Details on the maxima were given because it is precisely difficult to correctly identify them in Figure 12. We delete all values of maxima and kept only the information on their altitude in order to simplify the text. * 20: We replaced "convective surges positions" by "convective surge positions".

1-2: over a vertical distance varying * 1-2: We replaced "on a height varying" by "over a vertical distance varying".

6: in this case probably positive leaders themselves because of their very low power. 7: cloud-to-ground 9: intracloud 11: slightly tilted 12: does the positive leader really grow upward from the negative leader altitude? The graph suggests not (unclear). The vertical channel is mostly a later downward negative leader. On top of. 23: the slopes are missing in Fig 14 27: rather fast... This is still quite normal speed. See also van der Velde et al. 2014 27: 1.4x105 use superscript 28: Then, sources associated with the positive leader branch... 35: superscript * 6: Instead of deleting "(actually negative recoil leaders)", we prefer to develop further the possibility that observed positive leaders could in fact be the signature of retrograde negative breakdowns located close to

the tips of positive leaders. As analyzed by Edens et al. (2012) positive breakdowns do produce weak VHF emissions, however they may be masked by much stronger concurrent VHF emissions from negative breakdowns. But whatever positive leaders are directly or indirectly - via negative retrograde breakdowns at their tips – detected, this possibility does not question their presence and their location. * 7: We replaced "Cloud-to-Ground" by "cloud-to-ground". * 9: We replaced "intra-Cloud" by "intra-cloud". * 11: We replaced "slowly" by "slightly ". * 12: Yes it does, this is what we can observe with a low speed animation (XLMA). Only the upper end of this positive leader can be seen (indicated in green in the top panel of the bottom right graph in Figure 13). We mention that in the text. Yes later around the final phase of the flash the leader is a downward negative one (red lines in the upper panel of top left graph in Figure 13). Those negative leaders seem to start from the extreme southern tip of the upper positive charge layer. * We replace "on top the" by "on top of the". * 23: Figure 14 has been modified, the slopes are now present in right graph of Figure 14 (c.f. Specific comment #3 above). * 27: Yes, we now compare this value to that of van der Velde et al. (2014). * 27 and 35: Superscript has been correctly used. * 28: We corrected accordingly.

4: descends along... and propagates... into 6: nice to see that! It could be interesting to select this point as the origin of an alternative time-distance plot. See also van der Velde et al (2014) * 4: We modified the sentence according to the suggestion. * 6: The paper is already very long so that adding a new figure that deserves much more comments is probably not adequate. However this is a very good idea that we'll try to carry out in forthcoming studies.

2: needs rewriting, as one answer is also the question 7: van der Velde et al 2014, Lang et al 2010, Lu et al 2013, Soula et al 2015 could also fit 29: Could... not ? 33: one may wonder * 2: Yes, the question has been removed. * 7: We added these references,

except that of Lu et al. (2013) which is not available for us at the time we correct the manuscript. * 29: We corrected the interrogative sentence. * 33: We replaced "can" by "may".

5: Similarly/Additionally... * 5: The correction is OK, it has been proposed by both Referees

1: geometric performance 5: This performance 16: 2014-2016; a preliminary climatology * 1: We replaced "performances" by "geometric performance". * 5: We replaced "potential" by "geometric performance". * 16: The dates are corrected. We replaced "short of " by " preliminary".

5: in van der Velde and Montanya 2013 a negative leader develops vertically from a positive leader, it seems here we have the positive equivalent. Interesting. 7-8: see provided references with similar discharges * 5: Indeed it's a very interesting remark from Referee #1. Collaboration could probably take place with Oscar van der Velde and Joan Montanyà in a close future. * 7-8: We thank Referee #1 for sharing his knowledge. We added the references in section 4.2 (as said above in response to the specific comment #3) and changed the sentence of line 7-8.

Please also note the supplement to this comment:
https://www.atmos-meas-tech-discuss.net/amt-2019-192/amt-2019-192-AC1-supplement.pdf
* * *
[Figure]

![SAETTA LMA on 01 Oct 2015 — multi-panel plot showing aircraft track in altitude-time, altitude-distance, plan view, and altitude histogram with colored VHF source data]

**Fig. 1.** Track of a commercial aircraft flying at about 9 km altitude that produced an exceptionally huge stream of sparks, which may be distinguished in the cumulative number of filtered VHF sources in the to

---

## Author Comment (AC2) · 16 Sep 2019

Saturday, 14 September 2019

ATMOSPHERIC MEASUREMENT TECHNIQUES DISCUSSIONS

Manuscript amt-2019-192

Title: "SAETTA: high resolution 3D mapping of the total lightning activity in the Mediterranean basin over Corsica, with a focus on a MCS event"

Authors: Sylvain Coquillat, Eric Defer, Pierre de Guibert, Dominique Lambert, Jean-Pierre Pinty, Véronique Pont, Serge Prieur, Ronald J. Thomas, Paul R. Krehbiel,

[Figure]

William Rison

Dear Associate Editor,

We are very grateful to Referee #2 for his criticisms and suggestions that we tried to take into account to improve the manuscript significantly. We also thank Referee #2 for the comprehensive corrections he provided in the PDF scanned copy he sent. You will find below our item-by-item response (indicated by *) to the comments and recommendations of Referee #2. The proposed modifications appear in red font in the revised manuscript in order to readily identify them.

1. The "tri-level" (stacked?) flashes that were found to develop and then propagate in the trailing stratiform region may not be well-explained by conditions leading to the ice-based charging mechanism suggested by Dye and Bansemer (2019). The authors' discussion between line 25 on page 25 and line 7 on the following page describe other factors that require more-detailed analyses before interactions between separate-but overlapping storms and upper-level screening can be excluded. The vertical separation of the two upper charge regions associated with the upper-level flashes (I think that I see about 4 km in Figure 13 – not the "2-3 km" that they indicate on Page 25, line 22) seems rather large for screening, and rather and high for charge separation by mesoscale updrafts. Overall – my only suggestion is that the authors "soften" the statements in the abstract (lines 26-27 in page 1) and conclusions (lines 8-10 on page 25), instead of the current rather-direct attribution to this effect.

* We agree with Referee #2. We changed lines 21-26 in page 25 of the initial manuscript and also the abstract accordingly.

2. The description and performance characterization of the LMA network in Section 2 is excellent, but I have two issues that are worth mentioning. First, on lines 1-2 on page 4, the authors state a single benefit of having 12 stations rather than fewer stations. There are other good reasons worth mentioning, such as (1) redundancy/reliability (shortterm and long-term failure), (2) the effect of localized high-rate storms on a sensor's contribution to more-distant activity, and (3) the improved geometry for geo-location of distant lighting while maintaining height accuracy for nearby low-altitude lightning channels. I am sure that the authors can think of other benefits. The second issue relates to the depiction of vertical accuracy only at 10 km altitude. The vertical accuracy will be worse at about 3 km height (MSL) – above all the mountains. Users of these data would benefit from understanding this issue, either through additional figures or (at least) some words by the authors.

* Referee #2 is right on the 2 issues raised. It is well worth noticing most of the advantages arising from a 12-station network. We added the advantages proposed by Referee #2 and somewhat changed the initial sentence. We agree with Referee #2 about the second issue. We accordingly added some comments at the end of the 2nd paragraph of Section 2.3.

3. It would be nice if the first use of the xlma plots (Figure 5 for density and figure 6 for sources) were described in a bit more detail, including the distance (rather than lat-lon) scales, and then attributed to Ron Thomas and his xlma program.

* We added a sentence at the beginning of the first paragraph in Section 3.1 in order to describe the XLMA plots (see also the last sentence of the introduction about the XLMA tool and its attribution to Ron Thomas). As far as scales are considered, all XLMA plots are in Cartesian coordinates except Figure 5 left. As a matter of fact, Figure 5 displays 2 graphs. The left one is in lat-lon coordinates, the second one is in km-km coordinates. This allows comparing both sets of coordinates at least one time in the paper and identifying the latitude and longitude of the domain. We prefer to keep this information.

4. Most of the figures will be difficult for the reader to interpret. The fonts are too small to be read; there is wasted space (large separation between panels) that should be filled with real content (e.g., Figures 7 and 8, among others) ; It is difficult to see the overshooting-tops in Figure 10; the color separation in Figure 9 makes it difficult

to see the different years; and the "full xlma figures" maintain a lot of content (wasted visual space) that is not central to that points discussed by the authors (Figures 11 and 13). Maybe all figures should be reviewed by an author or colleague that never got the chance to manually zoom-in on the key features, so that they have the same disadvantages as the future readers.

\* We enlarged all the figures including XLMA plots, enlarged graphs in Figures 7 and 8 (and also in Figures 12 and 14), and changed colors and font size in Figure 9. We prefer to keep the whole XLMA panels in Figures 11 and 13 because they allow having good landmarks for example when analyzing Figure 14 left with the help of bottom left panel in top right Figure 13. As far as Figure 10 is considered, it is difficult to obtain a good quality image of its height-versus-time panel because the event is very long so that, for instance, overshooting-tops (or convective surges) cannot be correctly detailed. This is why we added information in the caption of Figure 11, which exhibits the detail of one of the convective surges (#5). If you think that the font size of all XLMA plots should be enlarged for the edition of the paper, we can provide new figures.

5. The terrain blockage analysis is an important element of this work. The technical discussion in the appendix does not mention how the authors handled refraction at VHF. Was it ignored, or was the radius of the earth adjusted (increased) to provide a simple correction for this? It would be interesting to know if sources were actually located at lower-than-expected heights.

\* For simplification, the atmospheric refraction is not taken into account in the calculation. This is now clearly stated in the text (Section 2.2) and in Appendix A, as rightly suggested by Referee #2. Inferences of this simplification on the results are drawn at the end of Section 2.2 (i.e. VHF sources can be detected even below the limits indicated in Figure 2).

6. The end date for the climatology (2016) seems odd, given the availability of data for 2017 and 2018. A sentence providing a rationale would be helpful. It might help

[Figure]

explain the awkward statement on line 15 of page 15 ("easy to extract") regarding the 2017 storm-day data.

* Only data from 2014 to 2016 have been fully processed. However, as it was easy and simple to get the information about the dates of the 2017 events (from quicklooks), we choose to add this information in Figure 9 in order to enrich the statistics on this point. This is now properly mentioned in the text.

7. There are several papers in the reference list that are not cited in the paper.

* Yes, we identified them and removed them.

8. I have a number of lesser comments, minor corrections, and editorial suggestions in a hand-annotated version of the manuscript. So that this review can be timely, a PDF scanned copy of this annotated version will be provided separately.

* We carefully corrected the manuscript according to the annotations of Referee #2. Two of them were somewhat difficult to read and/or understand (page 2 between lines 14 and 15; and page 13 in the right margin between lines 15 and the bottom of the page) so we did not consider them. Here are some comments about the corrections made, according to the pages of the pdf copy provided by Referee #2.

* Page 2: We kept the separation between "convection" and "latent heat release" because we describe the chain of processes one after the other (literally, convection is the vertical macroscopic motion, at the origin of the cooling of air, itself at the origin of water vapor condensation, itself at the origin of latent heat release...). OK for all other corrections.

* Page 3: OK for all corrections.

* Page 4: We added a comment about the time accuracy in order to clarify the purpose. OK for all other corrections.

* Pages 5 and 6: OK for the corrections.

* Page 8: We added a reference to Figure 13 in Thomas et al (2004) in order to clarify the evolution of the altitude z uncertainty versus r2. OK for all other corrections.

* Page 9: OK for the correction.

* Page 10: OK for the corrections. Data for the year 2017 are not considered because they have not yet been fully processed. Simple information such the number of lightning days is though available (and therefore reported in the paper) since it is issued from the daily quicklooks and not from the data processing. This is explained in Section 3.2.

* Page 11: OK for the corrections. We did not add the CG strokes data from the Euclid/Meteorage network because the display in Figure 6 left is not ambiguous about the cloud-to-ground nature of the lightning.

* Page 12: At the line 15 we did not mention the simultaneous propagation in the main negative charge region because it does not appears in the altitude versus East-West distance display in Figure 6 right (almost no red points distributed along the corresponding horizontal channel). Figure 6, as many other figures in the manuscript have been enlarged. OK for the corrections.

* Page 13: OK for the corrections (except the comments in the right margin as indicated above). We deleted the sentences relative to the aircraft trajectories in order to avoid the addition of new figures since the paper is already long. In fact this point does not really matter in this article. It will be later if we can get the flight data to evaluate the location efficiency of SAETTA. For information, the flight we were talking about produced an exceptionally huge amount of VHF sources (406210 sources, see figure here after).

* Page 14: OK for the corrections. We deleted the mention to the "secondary relief". The graphs in Figure7 have been enlarged.

* Page 15: We added a sentence to clarify the point raised about 2017 data. OK for the corrections.

* Page 16: OK for the corrections.

* Page 17: OK for the corrections. The high altitude discharges are said high but not very high. Referee #2 is right in his comment when saying that the considered discharges are not so high since they can appear at much higher altitudes as reported for instance by Krehbiel et al (2002). This is why we call them high altitude discharges rather than very high altitude discharges. As suggested, we added a sentence to describe the convective surges, with a reference to the paper of Krehbiel et al. (2002). We also changed the end of the 1st paragraph of Section 4.1 in order to clarify the comparison between present MCS and usual MCSs described in the literature.

* Page 18: OK for the corrections. The use of the XLMA tool is now mentioned at the end of the introduction.

* Page 19: Figures now enlarged, color scale not changed.

* Page 20: OK for the corrections. We modified the end of the sentence in line 9.

* Page 21: OK for the corrections. We enlarged the figures and also added some comments to help the reader. Details on the maxima were given because it is precisely difficult to correctly identify them in Figure 12. We simplified the corresponding sentences.

* Page 22: OK for the correction. We enlarged the graphs in Figure 13 instead of deleting some panels.

* Pages 23 and 24: OK for the corrections.

* Page 25: OK for the corrections. We modified the discussion about the conceptual model of Dye and Bansemer (2018, 2019) as suggested by Referee #2.

* Pages 26, 27, and 28: OK for the corrections.

* Page 29: Referee #2 is right about the atmospheric refraction. We neglected its effects in the calculation. It is now indicated in the text with a short analysis on the

consequences on the altitudes calculated.

* Page 31: OK for the corrections.

Please also note the supplement to this comment:
https://www.atmos-meas-tech-discuss.net/amt-2019-192/amt-2019-192-AC2-supplement.pdf
* * *
[Figure]

[Figure]

**Fig. 1.** Track of a commercial aircraft flying at about 9 km altitude that produced an exceptionally huge stream of sparks, which may be distinguished in the 
[revised manuscript text omitted]

Edens, H. E., K. B. Eack, E. M. Eastvedt, J. J. Trueblood, W. P. Winn, P. R. Krehbiel, G. D. Aulich, S. J. Hunyady, W. C. Murray, W. Rison, S. A. Behnke, and R. J. Thomas: VHF lightning mapping observations of a triggered lightning flash, Geophys. Res. Let., Vol. 39, L19807, doi:10.1029/2012GL053666, 2012.

Ely, B. L., Orville, R. E., Carey, L. D., and Hodapp, C. L.: Evolution of the total lightning structure in a leading-line, trailing-stratiform mesoscale convective system over Houston, Texas, J. Geophys. Res., 113, D08114, doi:10.1029/2007JD008445, 2008.

Emersic, C., P. L. Heinselman, D. R. MacGorman, and E. C. Bruning: Lightning activity in a hail producing storm observed with phased array radar, Mon. Weather Rev., 139, 1809–1825, doi:10.1175/2010MWR3574.1, 2011, 2001.

Fuchs, B. R., Rutledge, S. A., Bruning, E. C., Pierce, J. R., Kodros, J. K., Lang, T. J., MacGorman, D. R., Krehbiel, P. R., and Rison, W.: Environmental controls on storm intensity and charge structure in multiple regions of the continental United States, J. Geophys. Res. Atmos., 120, 6575-6596, doi:10.1002/2015JD023271, 2015.

Giorgi, F.: Climate change hot-spots, Geophys. Res. Let., Vol. 33, L08707, doi:10.1029/2006GL025734, 2006.

5     Houze Jr., R. A.: Cloud Dynamics, Academic, San Diego, p. 573, 1993.

Israel, H.: Atmospheric Electricity, Vol. II (translated from German), published by the National Science Foundation, Washington, DC by the Israel Program for Scientific Translations, 1973.

Kasemir, H. W.: Qualitative Ubersicht uber Potential-, Feld- und Ladungsverhaltnisse bei einer Blitzentladung in der Gewitterwolke (Qualitative Survey of the Potential, Field and Charge Conditions during a Lightning discharge in the

10    Thunderstorm Cloud), in: H. Israel (Ed.), Das Gewitter, Leipzig, Akadem. Verlagsgesellschaft, 1950.

Koshak, W. J., et al.: North Alabama Lightning Mapping Array (LMA): VHF source retrieval algorithm and error analyses, J. Atmos. Oceanic Technol., 21, 543– 558, 2004.

Krehbiel, P., T. Hamlin, Y. Zhang, J. Harlin, R. Thomas, and W. Rison, Three-dimensional total lightning observations with the lightning mapping array, 2002 International Lightning Detection Conference, Tucson, AZ, October 16-18, 2002.

15    Kuhlman, K. M., MacGorman, D. R., Biggerstaff, M. I., and Krehbiel, P. R.: Lightning initiation in the anvils of two supercell storms, Geophys. Res. Lett., 36, L07802, doi:10.1029/2008GL036650, 2009.

Lambert, D., and Argence, S.: Preliminary study of an intense rainfall episode in Corsica, 14 September 2006. Adv. Geosci., 16, 125–129, 2008.

Lang, T., Miller, L. J., Weisman, M., Rutledge, S. A., Barker III, L. J., Bringi, V. N., Chandrasekar, V., Detwiler, A.,

20    Doesken, N., Helsdon, J., Knight, C., Krehbiel, P., Lyons, W., MacGorman, D., Rasmussen, E., Rison, W., Rust, W. D., and Thomas, R. J.: The Severe Thunderstorm Electrification and Precipitation Study (STEPS), Bull. Amer. Meteor. Soc., 85, 1107–1125, 2004.

Lang, T. J., W. A. Lyons, S. A. Rutledge, J. D. Meyer, D. R. MacGorman, and S. A. Cummer: Transient luminous events above two mesoscale convective systems: Storm structure and evolution, J. Geophys. Res., 115, A00E22,

25    doi:10.1029/2009JA014500, 2010.

Lang, T. J., J. Li, W. A. Lyons, S. A. Cummer, S. A. Rutledge, and D. R. MacGorman: Transient luminous events above two mesoscale convective systems: Charge moment change analysis, J. Geophys. Res., 116, A10306, doi:10.1029/2011JA016758, 2011.

[revised manuscript text omitted]

Ushio, T., S. J. Heckman, H. J. Christian, and Z. I. Kawasaki: Vertical development of lightning activity observed by the LDAR system: Lightning bubbles. J. Appl. Meteor., 42, 165–174, 2003.

5    van der Velde, O. A., and Montanyà, J.: Asymmetries in bidirectional leader development of lightning flashes, J. Geophys. Res. Atmos., 118, 1–16, doi:10.1002/2013JD020257, 2013.

van der Velde, O. A., J. Montanyà, S. Soula, N. Pineda, and J. Mlynarczyk: Bidirectional leader development in sprite-producing positive cloud-to-ground flashes: Origins and characteristics of positive and negative leaders, J. Geophys. Res. Atmos., 119, 12,755–12,779, doi:10.1002/2013JD021291, 2014.

10   Weiss, S., D. R. MacGorman, and K. M. Calhoun: Lightning in the Anvils of Supercell Thunderstorms, Monthly Weather Revue, Vol 40, 2064-2079, DOI: 10.1175/MWR-D-11-00312.1, 2012.

---

## Author Comment (AC3) · 16 Sep 2019

The corrections are mentioned in the response to Referee #2
* * *

---

## Author Comment (AC4) · 25 Sep 2019

Wednesday, 25 September 2019

ATMOSPHERIC MEASUREMENT TECHNIQUES DISCUSSIONS

Manuscript amt-2019-192

Title: "SAETTA: high resolution 3D mapping of the total lightning activity in the Mediterranean basin over Corsica, with a focus on a MCS event"

Authors: Sylvain Coquillat, Eric Defer, Pierre de Guibert, Dominique Lambert, Jean-Pierre Pinty, Véronique Pont, Serge Prieur, Ronald J. Thomas, Paul R. Krehbiel,

[Figure]

William Rison

Dear Associate Editor,

We thank you for having accepted our study for publication in AMT. We made the technical corrections that you requested (see below our point-by-point response). In particular, we checked - and modified where needed - all references. The modifications appear in red font in the new manuscript. Some figures have been moved so that the layout does not separate these figures from their legend.

Sincerely yours,

Sylvain Coquillat

P1, L12: resolutions –> resolution

Corrected in page 1.

P2, L12: please explain "LF/VLF" and later on "VHF"

Explained in page 2.

P4, L 16: "12 ns rms"? Please clarify.

We deleted "rms", which was mentioned by mistake, page 4.

P7, L5: ...is expected TO allow FOR a better...

Corrected in page 7.

P12, L2: ... and therefore NEED TO (or HAVE TO?) be considered in the calculation...

Sentence corrected in page 12 by "may be considered".

P13, L24: I would not say that Corsica is representative of the convection in the whole Western Mediterranean basin. Please omit that part of the sentence or convince me otherwise.

[Figure]

Part of the sentence deleted in page 13, you are right.

P22, Fig. 12: Please clarify the size definitions: What is a big flash etc?

Clarification made in page 22 at the end of the figure caption.

P20, L32: virtuel accuracy –> vertical accuracy

Corrected in page 30.

References: Please check your bibliographic entries so that all of the citations are matched (citation in the text AND entry in reference list). E.g. Barthlott and Kirshbaum (2013) is missing in the reference list, Barthlott et al. (2014) –> Barthlott et al. (2016) + Vol + pages

We checked all references. We added some that were missing (Tessendorf; Wiens), and modified others where needed. The references are now corrected in the text (pages 3; 10; and 17) and in the references section (pages 35; 36; 37; 38; and 39).

Please also note the supplement to this comment:
https://www.atmos-meas-tech-discuss.net/amt-2019-192/amt-2019-192-AC4-supplement.pdf

**Supplement:**

[revised manuscript text omitted]

Houze Jr., R. A.: Cloud Dynamics, Academic, San Diego, p. 573, 1993.

Israel, H.: Atmospheric Electricity, Vol. II (translated from German), published by the National Science Foundation, Washington, DC by the Israel Program for Scientific Translations, 1973.

Kasemir, H. W.: Qualitative Ubersicht uber Potential-, Feld- und Ladungsverhaltnisse bei einer Blitzentladung in der Gewitterwolke (Qualitative Survey of the Potential, Field and Charge Conditions during a Lightning discharge in the Thunderstorm Cloud), in: H. Israel (Ed.), Das Gewitter, Leipzig, Akadem. Verlagsgesellschaft, 1950.

Koshak, W. J., et al.: North Alabama Lightning Mapping Array (LMA): VHF source retrieval algorithm and error analyses, J. Atmos. Oceanic Technol., 21, 543– 558, 2004.

Krehbiel, P., T. Hamlin, Y. Zhang, J. Harlin, R. Thomas, and W. Rison, Three-dimensional total lightning observations with the lightning mapping array, 2002 International Lightning Detection Conference, Tucson, AZ, October 16-18, 2002.

Kuhlman, K. M., MacGorman, D. R., Biggerstaff, M. I., and Krehbiel, P. R.: Lightning initiation in the anvils of two supercell storms, Geophys. Res. Lett., 36, L07802, doi:10.1029/2008GL036650, 2009.

Lambert, D., and Argence, S.: Preliminary study of an intense rainfall episode in Corsica, 14 September 2006. Adv. Geosci., 16, 125–129, doi.org/10.5194/adgeo-16-125-2008, 2008.

Lang, T., Miller, L. J., Weisman, M., Rutledge, S. A., Barker III, L. J., Bringi, V. N., Chandrasekar, V., Detwiler, A., Doesken, N., Helsdon, J., Knight, C., Krehbiel, P., Lyons, W., MacGorman, D., Rasmussen, E., Rison, W., Rust, W. D., and Thomas, R. J.: The Severe Thunderstorm Electrification and Precipitation Study (STEPS), Bull. Amer. Meteor. Soc., 85, 1107–1125, 2004.

Lang, T. J., W. A. Lyons, S. A. Rutledge, J. D. Meyer, D. R. MacGorman, and S. A. Cummer: Transient luminous events above two mesoscale convective systems: Storm structure and evolution, J. Geophys. Res., 115, A00E22, doi:10.1029/2009JA014500, 2010.

Lang, T. J., J. Li, W. A. Lyons, S. A. Cummer, S. A. Rutledge, and D. R. MacGorman: Transient luminous events above two mesoscale convective systems: Charge moment change analysis, J. Geophys. Res., 116, A10306, doi:10.1029/2011JA016758, 2011.

[revised manuscript text omitted]

15   Ushio, T., S. J. Heckman, H. J. Christian, and Z. I. Kawasaki: Vertical development of lightning activity observed by the LDAR system: Lightning bubbles. J. Appl. Meteor., 42, 165–174, 2003.

van der Velde, O. A., and Montanyà, J.: Asymmetries in bidirectional leader development of lightning flashes, J. Geophys. Res. Atmos., 118, 1–16, doi:10.1002/2013JD020257, 2013.

van der Velde, O. A., J. Montanyà, S. Soula, N. Pineda, and J. Mlynarczyk: Bidirectional leader development in sprite-

20   producing positive cloud-to-ground flashes: Origins and characteristics of positive and negative leaders, J. Geophys. Res. Atmos., 119, 12,755–12,779, doi:10.1002/2013JD021291, 2014.

Weiss, S., D. R. MacGorman, and K. M. Calhoun: Lightning in the Anvils of Supercell Thunderstorms, Monthly Weather Revue, Vol 40, 2064-2079, DOI: 10.1175/MWR-D-11-00312.1, 2012.

Wiens, K. C., Rutledge, S. A., and Tessendorf, S. A.: The 29 June 2000 Supercell Observed during STEPS. Part II:

25   Lightning and Charge Structure, J. Atmos. Sci., 62, 4151-4177, 2005.